# Sex- and age-specific burden of self-harm and suicide mortality: A national and subnational study in Iran

Sohrab Amiri  and Jannat Mashayekhi

Spiritual Health Research Center, Lifestyle Institute, Baqiyatallah University of Medical Sciences, Tehran, Iran

prevalence; incidence; self-harm; suicide mortality; global burden of disease

**Corresponding author:**
Sohrab Amiri;
Email: amirysohrab@yahoo.com

## Abstract

This study focuses on the national and subnational estimation of prevalence, incidence, disability-adjusted life years (DALYs) related to self-harm and suicide mortality in Iran. These indicators of disease burden were analyzed over the period from 1990 to 2021, with stratifications based on sex, age and geographic location. Additionally, the percentage change observed between 1990 and 2021 was documented. The age-standardized prevalence rate (per 100,000) of self-harm decreased from 173.92 (95% UI: 146.13–208.75) in 1990 to 131.2 (95% UI: 110.55–156.67) in 2021, reflecting a percentage change of −0.25% over the period. In terms of self-harm prevalence in 2021, males had a higher rate (137.62 per 100,000) compared to females (124.82 per 100,000). The findings of the current study revealed that, despite significant challenges such as demographic shifts, economic instability and the impacts of war, the trends in self-harm incidents and suicide mortality rates in Iran have generally been on the decline. Additionally, it was observed that suicide-related deaths were more prevalent among males when compared to females. However, when examining self-harm behaviors over previous decades, these acts appeared to be more frequent among females.

## Impact statement

One of the important factors in self-harm and suicide mortality is geographical differences, which lead to cultural and religious differences. Consequently, during these years, the nation grappled with a range of issues stemming from the aftermath of war, including an elevated incidence of mental health disorders, economic instability and pressures associated with international policies. Recognizing the significant role of mental health in enhancing overall community well-being, health policies should prioritize expanding access to mental health care services. This can be achieved through proactive initiatives that focus on education at various levels, including community programs, schools, universities and health centers. Efforts should aim to improve mental health literacy while addressing cases requiring psychological interventions, thereby reducing the economic and social impact of mental health issues.

## Introduction

Self-harm and suicide represent significant health and social challenges on a global scale (Knipe et al., 2022). The World Health Organization (WHO) conceptualizes self-harm as the act of intentionally causing injury to oneself as a means of managing or expressing severe emotional distress and internal conflict. While individuals engaging in such behavior typically do not aim to end their lives, the potential consequences can nonetheless be fatal. Common manifestations of self-harm include deliberate poisoning through the ingestion of an excessive amount of medication or harmful substances, self-inflicted wounds through cutting or burning, head-banging against solid objects or inflicting physical pain by punching or striking oneself against hard surfaces. Although suicidal intent is generally absent among those who engage in self-harm, the risks associated with these actions remain profoundly serious (World Health Organization, 2019). Suicide can be defined as "the act of intentionally carrying out an action to kill oneself" (World Health Organization, 2019).

The WHO estimates that over 700,000 individuals lose their lives to suicide annually worldwide (World Health Organization, 2023). In 2019, the global age-standardized suicide rate stood at 9.0 per 100,000 people, with men exhibiting a higher rate of 12.6 per 100,000 compared to women, who had a rate of 5.4 per 100,000 (World Health Organization, 2021). Although suicide mortality rates are more common in males, self-harm is more common in females (Knipe et al., 2022). Self-harm and suicide result from the interaction of complex factors and are influenced by variables such as age, sex, ethnicity and geography (Knipe et al., 2022).

The region encompassing North Africa and the Middle East, including Iran, has experienced significant challenges in recent decades, marked by rapid population growth, ongoing conflicts and persistent warfare. The nations within the Middle East exhibit substantial ethnic and demographic commonalities, as noted (Moradinazar et al., 2022). Islam serves as the predominant religion across these countries. While historically, the rate of suicide has generally been low in most Islamic societies, contemporary evidence suggests a concerning upward trajectory in this phenomenon (Pritchard and Amanullah, 2007; Mirhashemi et al., 2016). In Iran, the overall suicide rate is 5.3 per 100,000, with rates of 7 and 3.6 for males and females respectively (World Health Organization, 2014). Some determinants of suicide in Iran include family conflict, marital problems, economic constraints and educational failures (Nazarzadeh et al., 2013).

During the past decades, Iran has been affected by economic turmoil, war and international politics (Danaei et al., 2019). After the end of the war in 1988, Iran entered a period of construction in health and nonhealth infrastructures, which led to an increase in gross domestic product (Bank, 2021). Demographic transition has led to population growth, a change in fertility, increased life expectancy, aging and epidemiological transition (Farzadfar et al., 2022; Bhattacharjee et al., 2024; GBD 2021 Causes of Death Collaborators, 2024). Access to mental health care services has not been developed in response to the growth of mental disorders (Farzadfar et al., 2022). The prevalence of mental health issues has gained considerable attention since the conclusion of the Iran–Iraq war. This concern is further underscored by the findings of the Iranian Mental Health Survey, which indicates a substantial occurrence of psychiatric disorders within the country (Sharifi et al., 2015; Danaei et al., 2019). This study focuses on estimating the prevalence, incidence and disability-adjusted life years (DALYs) associated with self-harm and suicide mortality in Iran at both national and subnational levels, using data from the Global Burden of Disease (GBD) 2021.

## Methods

### Protocol

This manuscript was produced as part of the GBD Collaborator Network and following the GBD Protocol.

### Data source

This research was based on the GBD 2021 (Ferrari et al., 2024). The extracted burden of disease indicators included prevalence, incidence, DALYs, years lived with disability (YLDs), years of life lost (YLLs) and death for 371 diseases and injuries, along with estimates of healthy life expectancy. These estimates are provided for sex and age groups, and for 204 countries and territories, including subnational estimates for 21 countries (Ferrari et al., 2024). Data sources used in GBD 2021 included 100,983 data sources (19,189 new data sources for DALYs), 12 new causes and other important methodological updates (Ferrari et al., 2024). These indicators were examined at the national and subnational levels in Iran. More details of GBD 2021 are presented elsewhere (Ferrari et al., 2024).

### Case definitions

Self-harm in GBD 2021 is "deliberate bodily damage inflicted on oneself resulting in death or injury. ICD-9: E950-E959; ICD-10: X60-X64.9, X66-X84.9, Y87.0" (2021, 2024; Ferrari et al., 2024). This contains two subclasses: (i) Self-harm by firearm, defined as "Death or disability inflicted by the intentional use of a firearm on oneself. ICD-9: E955-E955.9; ICD-10: X72-X74.9" (2021, 2024; Ferrari et al., 2024); and (ii) Self-harm by other specified means, defined as "Death or occurrence of deliberate bodily damage inflicted on oneself resulting in death by means of self-poisoning, medication overdose, transport, falling from height, hanging or strangulation or other mechanisms not including firearms. ICD9: E950-E954, E956-E959; ICD10: X60-X64.9, X66-X67.9, X69-X71.9, X75-X75.9, X77-X84.9, Y87.0" (2021, 2024; Ferrari et al., 2024).

### Estimation framework

YLDs were calculated "with a microsimulation process that used estimated age-sex-location-year-specific prevalent counts of nonfatal disease sequelae (consequences of a disease or injury) for each cause and disability weights for each sequela as the input estimates at varying levels of severity by an appropriate disability weight" (Ferrari et al., 2024). YLLs were calculated as "the product of estimated age-sex-location-year-specific deaths and the standard life expectancy at the age death occurred for a given cause" (Ferrari et al., 2024). DALYs were calculated as the sum of YLDs and YLLs (Ferrari et al., 2024).

### Statistics

All-age count estimates and age-standardized rate prevalence (per 100,000) were calculated for prevalence, incidence, DALYs and death. Each of the disease burden indicators was examined in the period of 1990–2021, stratified by sex, age and location, and the % change between 1990 and 2021 was reported. The 95% uncertainty interval (UI) was reported for each of the reported estimates. More details about data, data processing and modeling are provided elsewhere and are related to GBD 2021 (Ferrari et al., 2024). GBD 2021 complies with the Guidelines for Accurate and Transparent Health Estimates Reporting (Stevens et al., 2016) and analyses were completed using Python (version 3.10.4), Stata (version 13.1) and R (version 4.2.1).

## Results

### Prevalence of self-harm in Iran from 1990 to 2021

Age-standardized rate prevalence (per 100,000) of self-harm in 1990 was 173.92 (95% UI: 146.13–208.75) versus 131.2 (95% UI: 110.55–156.67) in 2021, thus indicating a decrease; the percentage change from 1990 to 2021 was −25%. All-ages count estimates of self-harm in 1990 were 65,834 (95% UI: 55,038–79,967) versus 122,781 (95% UI: 103,040–147,165) in 2021; the percentage change from 1990 to 2021 was 0.86%. While the age-standardized rate has shown a decline, the absolute count estimates have increased over recent decades, primarily due to population growth (Table 1 and Figure 1).

### Incidence of self-harm in Iran from 1990 to 2021

The age-standardized incidence rate of self-harm per 100,000 population was 57.34 (95% UI: 46.67–69.95) in 1990, compared to 46.01 (95% UI: 36.07–57.53) in 2021. This reflects a percentage

**Table 1.** All-ages counts and age-standardized rate (per 100,000) prevalence of self-harm in Iran, stratified by provinces, 1990–2021

| | Year | | | | | | | | |
|---|---|---|---|---|---|---|---|---|---|
| | 1990 | | | 2021 | | | Percentage change 1990–2021 | | |
| Location | Value | Lower | Upper | Value | Lower | Upper | Value | Lower | Upper |
| Age-standardized rate prevalence (per 100,000) | | | | | | | | | |
| Iran | 173.92 | 146.13 | 208.75 | 131.2 | 110.55 | 156.67 | −0.25 | −0.28 | −0.22 |
| Alborz | 178.5 | 149.19 | 214.24 | 152.64 | 127.58 | 181.88 | −0.14 | −0.17 | −0.12 |
| Ardebil | 203.51 | 171.14 | 243.77 | 139.42 | 117.74 | 166.11 | −0.31 | −0.35 | −0.28 |
| Bushehr | 184.88 | 155.03 | 221.06 | 125.95 | 107.24 | 149.5 | −0.32 | −0.36 | −0.28 |
| Chahar Mahaal and Bakhtiari | 212.45 | 177.76 | 253.55 | 132.24 | 110.95 | 157.88 | −0.38 | −0.41 | −0.35 |
| East Azarbayejan | 175.65 | 146.19 | 215.15 | 137.76 | 115.82 | 164.44 | −0.22 | −0.26 | −0.18 |
| Fars | 211.5 | 175.97 | 255.93 | 150.55 | 125.4 | 180.46 | −0.29 | −0.32 | −0.26 |
| Gilan | 158.59 | 131.6 | 191.18 | 133.92 | 112.27 | 159.34 | −0.16 | −0.19 | −0.12 |
| Golestan | 228.88 | 189.83 | 279.77 | 147.88 | 123.62 | 177.4 | −0.35 | −0.39 | −0.32 |
| Hamadan | 209.33 | 173.99 | 255.23 | 160.45 | 134.13 | 191 | −0.23 | −0.28 | −0.19 |
| Hormozgan | 165.42 | 139.31 | 196.97 | 126.22 | 106.7 | 151.08 | −0.24 | −0.27 | −0.21 |
| Ilam | 276.54 | 230.25 | 334.34 | 188.33 | 157.11 | 223.85 | −0.32 | −0.35 | −0.29 |
| Isfahan | 133.54 | 112.54 | 160.13 | 117.3 | 98.45 | 140.14 | −0.12 | −0.15 | −0.1 |
| Kerman | 165.18 | 137.85 | 199.46 | 131.3 | 110.3 | 157.52 | −0.21 | −0.24 | −0.18 |
| Kermanshah | 301.39 | 251.47 | 370.07 | 178.76 | 148.3 | 213.86 | −0.41 | −0.44 | −0.38 |
| Khorasan-e-Razavi | 169.05 | 141.02 | 204.91 | 125.02 | 105.77 | 148.83 | −0.26 | −0.3 | −0.22 |
| Khuzestan | 185.52 | 155.27 | 222.52 | 140.34 | 117.2 | 169.14 | −0.24 | −0.27 | −0.22 |
| Kohgiluyeh and Boyer-Ahmad | 226.6 | 188.32 | 278.46 | 169.1 | 140.05 | 203.16 | −0.25 | −0.29 | −0.22 |
| Kurdistan | 217.73 | 182.21 | 262.47 | 131.14 | 110.26 | 156.09 | −0.4 | −0.43 | −0.37 |
| Lorestan | 251.81 | 210.01 | 303.26 | 174.12 | 144.77 | 205.66 | −0.31 | −0.34 | −0.27 |
| Markazi | 162.87 | 136.18 | 196.35 | 114.55 | 96.64 | 136.53 | −0.3 | −0.33 | −0.26 |
| Mazandaran | 134.46 | 112.75 | 161.84 | 122.26 | 102.44 | 146.53 | −0.09 | −0.12 | −0.06 |
| North Khorasan | 205.6 | 172.25 | 250.6 | 140.01 | 117.22 | 167.14 | −0.32 | −0.35 | −0.29 |
| Qazvin | 122.68 | 103.93 | 146.4 | 116.23 | 98.39 | 138.28 | −0.05 | −0.09 | −0.02 |
| Qom | 123.74 | 105.3 | 147.14 | 108.72 | 92.02 | 129.28 | −0.12 | −0.15 | −0.09 |
| Semnan | 123.04 | 103.9 | 147.66 | 109.19 | 92.72 | 129.22 | −0.11 | −0.15 | −0.08 |
| Sistan and Baluchistan | 115.68 | 99.28 | 137.06 | 121.66 | 103.55 | 144.79 | 0.05 | 0.02 | 0.08 |
| South Khorasan | 138.54 | 117.01 | 165.56 | 122.12 | 103.87 | 146.15 | −0.12 | −0.16 | −0.09 |
| Tehran | 146.96 | 125.22 | 173.76 | 109.65 | 93.76 | 129.7 | −0.25 | −0.28 | −0.23 |
| West Azarbayejan | 209.73 | 174.08 | 258.42 | 155.46 | 129.71 | 185.95 | −0.26 | −0.3 | −0.22 |
| Yazd | 139.26 | 118.03 | 165.83 | 112.4 | 94.88 | 133.54 | −0.19 | −0.23 | −0.17 |
| Zanjan | 124.69 | 104.09 | 150.7 | 106.84 | 90.84 | 127.12 | −0.14 | −0.19 | −0.11 |
| All-ages counts estimates | | | | | | | | | |
| Iran | 65,834.87 | 55,038.75 | 79,967.61 | 122,781.76 | 103,040.92 | 147,165.99 | 0.86 | 0.75 | 0.97 |
| Alborz | 1,696.82 | 1,405.40 | 2,081.74 | 5,259.88 | 4,377.23 | 6,314.42 | 2.1 | 1.93 | 2.27 |
| Ardebil | 1,436.58 | 1,198.00 | 1,739.51 | 2,019.58 | 1,696.22 | 2,413.46 | 0.41 | 0.31 | 0.5 |
| Bushehr | 804.56 | 667.29 | 983.66 | 1,573.10 | 1,329.39 | 1,881.32 | 0.96 | 0.81 | 1.08 |
| Chahar Mahaal and Bakhtiari | 914.4 | 760.57 | 1,108.48 | 1,413.89 | 1,179.86 | 1,700.93 | 0.55 | 0.44 | 0.64 |
| East Azarbayejan | 4,103.35 | 3,397.43 | 5,076.82 | 6,465.85 | 5,421.25 | 7,712.64 | 0.58 | 0.45 | 0.69 |
| Fars | 4,935.51 | 4,059.97 | 6,077.20 | 8,718.66 | 7,230.23 | 10,491.51 | 0.77 | 0.65 | 0.88 |

*(Continued)*

**Table 1.** (*Continued*)

| Location | 1990 | | | 2021 | | | Percentage change 1990–2021 | | |
|---|---|---|---|---|---|---|---|---|---|
| | Value | Lower | Upper | Value | Lower | Upper | Value | Lower | Upper |
| Gilan | 2,643.18 | 2,187.74 | 3,231.66 | 4,440.05 | 3,727.77 | 5,293.25 | 0.68 | 0.58 | 0.79 |
| Golestan | 1,899.27 | 1,564.14 | 2,373.57 | 3,029.73 | 2,520.14 | 3,653.02 | 0.6 | 0.46 | 0.71 |
| Hamadan | 2,341.79 | 1,934.59 | 2,891.69 | 3,232.73 | 2,694.26 | 3,851.66 | 0.38 | 0.27 | 0.48 |
| Hormozgan | 934.97 | 782.04 | 1,127.97 | 2,362.41 | 1,971.94 | 2,852.77 | 1.53 | 1.39 | 1.63 |
| Ilam | 714.73 | 584.95 | 884.35 | 1,267.53 | 1,049.95 | 1,523.53 | 0.77 | 0.64 | 0.89 |
| Isfahan | 3,447.57 | 2,892.63 | 4,196.77 | 7,361.04 | 6,172.07 | 8,820.49 | 1.14 | 1.01 | 1.25 |
| Kerman | 1,943.91 | 1,620.02 | 2,375.54 | 4,561.80 | 3,801.49 | 5,503.58 | 1.35 | 1.22 | 1.47 |
| Kermanshah | 3,285.21 | 2,701.59 | 4,109.75 | 4,128.72 | 3,416.37 | 4,953.89 | 0.26 | 0.16 | 0.34 |
| Khorasan-e-Razavi | 5,164.58 | 4,300.56 | 6,309.27 | 8,880.43 | 7,477.54 | 10,626.20 | 0.72 | 0.59 | 0.83 |
| Khuzestan | 3,535.05 | 2,942.60 | 4,346.78 | 7,056.76 | 5,847.49 | 8,562.09 | 1 | 0.9 | 1.09 |
| Kohgiluyeh and Boyer-Ahmad | 635.97 | 521.02 | 801.32 | 1,333.77 | 1,093.08 | 1,631.27 | 1.1 | 0.95 | 1.23 |
| Kurdistan | 1,726.69 | 1,432.93 | 2,120.29 | 2,496.20 | 2,088.58 | 2,990.30 | 0.45 | 0.34 | 0.54 |
| Lorestan | 2,365.37 | 1,961.41 | 2,913.33 | 3,347.92 | 2,773.13 | 3,989.78 | 0.42 | 0.32 | 0.51 |
| Markazi | 1,327.51 | 1,111.45 | 1,616.79 | 1,950.96 | 1,643.72 | 2,325.07 | 0.47 | 0.36 | 0.57 |
| Mazandaran | 2,368.55 | 1,972.35 | 2,909.53 | 5,241.43 | 4,394.54 | 6,269.77 | 1.21 | 1.09 | 1.35 |
| North Khorasan | 855.31 | 712.79 | 1,061.82 | 1,256.70 | 1,047.33 | 1,508.13 | 0.47 | 0.36 | 0.55 |
| Qazvin | 735.99 | 617.77 | 893.05 | 1,748.59 | 1,475.38 | 2,088.49 | 1.38 | 1.24 | 1.51 |
| Qom | 578.08 | 487.18 | 702.7 | 1,618.84 | 1,361.07 | 1,939.42 | 1.8 | 1.64 | 1.97 |
| Semnan | 423.16 | 356.45 | 510.08 | 945.74 | 799.38 | 1,128.53 | 1.23 | 1.13 | 1.33 |
| Sistan and Baluchistan | 967.64 | 822.49 | 1,157.74 | 2,888.69 | 2,427.41 | 3,464.85 | 1.99 | 1.88 | 2.09 |
| South Khorasan | 614.73 | 519.05 | 739.81 | 1,076.21 | 911.1 | 1,291.98 | 0.75 | 0.66 | 0.83 |
| Tehran | 8,903.83 | 7,516.98 | 10,694.68 | 18,736.82 | 15,979.39 | 22,234.83 | 1.1 | 0.99 | 1.2 |
| West Azarbayejan | 3,193.78 | 2,621.41 | 4,007.53 | 5,687.11 | 4,723.61 | 6,828.00 | 0.78 | 0.65 | 0.9 |
| Yazd | 635.95 | 536.06 | 769.85 | 1,381.60 | 1,160.92 | 1,651.76 | 1.17 | 1.06 | 1.28 |
| Zanjan | 700.84 | 583.61 | 860.52 | 1,299.00 | 1,099.52 | 1,550.85 | 0.85 | 0.74 | 0.96 |

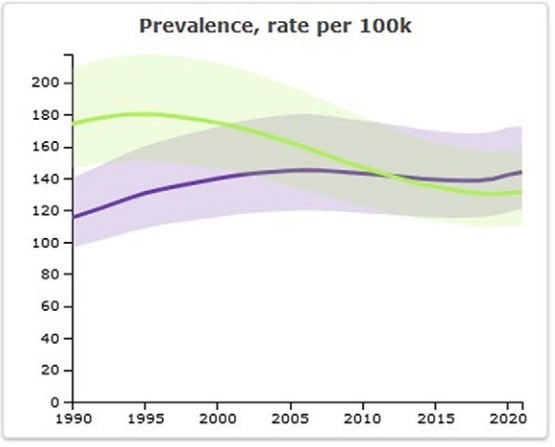

**Figure 1.** Trend in prevalence of self-harm in Iran, 1990–2021.

change of −0.20% over the period, indicating a downward trend. All-ages count estimates increased from 32,083.13 (95% UI: 25,531–40,402) in 1990 to 40,908.57 (95% UI: 32,066.60–50,434.56) in 2021; the percentage change from 1990 to 2021 was 0.28%. Between 2000 and 2010, the estimated incidence of self-harm showed an overall increase, peaking in 2005. Nevertheless, when adjusted for age, the incidence rates indicate a declining trend (Table 2 and Figure 2).

**Table 2.** All-ages counts and age-standardized rate (per 100,000) incidence of self-harm in Iran, stratified by provinces, 1990–2021

| | Year | | | | | | | | |
| | 1990 | | | 2021 | | | Percentage change 1990–2021 | | |
| Location | Value | Lower | Upper | Value | Lower | Upper | Value | Lower | Upper |
|---|---|---|---|---|---|---|---|---|---|
| Age-standardized rate incidence (per 100,000) | | | | | | | | | |
| Iran | 57.34 | 46.67 | 69.95 | 46.01 | 36.07 | 57.53 | −0.2 | −0.26 | −0.15 |
| Alborz | 58.75 | 47.85 | 71.94 | 54.12 | 42.58 | 67.02 | −0.08 | −0.15 | −0.01 |
| Ardebil | 66.26 | 53.35 | 81.24 | 48.96 | 38.34 | 61.13 | −0.26 | −0.32 | −0.2 |
| Bushehr | 62.26 | 50.4 | 76.03 | 45.3 | 35.66 | 56.3 | −0.27 | −0.34 | −0.21 |
| Chahar Mahaal and Bakhtiari | 71.82 | 58.3 | 88.23 | 48.59 | 38.34 | 61.09 | −0.32 | −0.38 | −0.27 |
| East Azarbayejan | 55.48 | 44.64 | 68.17 | 47.45 | 36.97 | 59.62 | −0.14 | −0.22 | −0.06 |
| Fars | 69.82 | 56.66 | 85.59 | 52.91 | 41.66 | 66.62 | −0.24 | −0.3 | −0.18 |
| Gilan | 55.97 | 45.52 | 68.17 | 47.72 | 37.45 | 59.53 | −0.15 | −0.23 | −0.08 |
| Golestan | 75.49 | 60.77 | 92.1 | 51.35 | 40.35 | 63.55 | −0.32 | −0.38 | −0.26 |
| Hamadan | 66.09 | 53.63 | 80.8 | 55.26 | 42.9 | 68.57 | −0.16 | −0.24 | −0.09 |
| Hormozgan | 52.5 | 42.54 | 64.54 | 42.75 | 33.5 | 53.47 | −0.19 | −0.25 | −0.11 |
| Ilam | 94.06 | 76.65 | 114.12 | 67.71 | 53.48 | 83.73 | −0.28 | −0.34 | −0.22 |
| Isfahan | 43.51 | 35.1 | 53.68 | 41 | 32.37 | 51.27 | −0.06 | −0.13 | 0.01 |
| Kerman | 52.94 | 42.66 | 65.05 | 44.85 | 34.99 | 56.4 | −0.15 | −0.23 | −0.07 |
| Kermanshah | 98.99 | 81.37 | 121.18 | 62.73 | 49.26 | 77.69 | −0.37 | −0.42 | −0.31 |
| Khorasan-e-Razavi | 54.11 | 43.74 | 67.21 | 42.95 | 33.59 | 53.64 | −0.21 | −0.27 | −0.14 |
| Khuzestan | 61.47 | 49.24 | 75.94 | 49.07 | 38.3 | 61.32 | −0.2 | −0.26 | −0.15 |
| Kohgiluyeh and Boyer-Ahmad | 75.31 | 60.75 | 92.62 | 60.5 | 47.55 | 76.12 | −0.2 | −0.26 | −0.12 |
| Kurdistan | 71.16 | 57.59 | 86.6 | 45.7 | 35.79 | 57.71 | −0.36 | −0.42 | −0.3 |
| Lorestan | 83.76 | 67.77 | 102.28 | 63.28 | 49.55 | 78.07 | −0.24 | −0.3 | −0.19 |
| Markazi | 51.66 | 41.77 | 62.96 | 39.46 | 31.08 | 49.8 | −0.24 | −0.3 | −0.17 |
| Mazandaran | 44.86 | 36.35 | 54.98 | 43.56 | 34.06 | 54.22 | −0.03 | −0.1 | 0.05 |
| North Khorasan | 66.26 | 53.58 | 80.45 | 48.4 | 38.25 | 60.81 | −0.27 | −0.33 | −0.21 |
| Qazvin | 39.42 | 31.94 | 48.2 | 40.52 | 31.88 | 50.76 | 0.03 | −0.05 | 0.13 |
| Qom | 38.71 | 31.52 | 47.94 | 37.23 | 29.02 | 46.83 | −0.04 | −0.12 | 0.05 |
| Semnan | 39.16 | 31.78 | 48.14 | 38.01 | 29.58 | 47.79 | −0.03 | −0.11 | 0.04 |
| Sistan and Baluchistan | 36.32 | 29.18 | 44.62 | 39.48 | 30.79 | 50.01 | 0.09 | 0.02 | 0.16 |
| South Khorasan | 44.36 | 36.08 | 54.6 | 42.09 | 33.09 | 53.21 | −0.05 | −0.13 | 0.03 |
| Tehran | 48.8 | 39.64 | 59.27 | 39.32 | 30.69 | 49.46 | −0.19 | −0.26 | −0.14 |
| West Azarbayejan | 67.54 | 54.44 | 83.27 | 54.2 | 42.62 | 67.46 | −0.2 | −0.26 | −0.13 |
| Yazd | 45.25 | 36.24 | 55.14 | 39.36 | 30.72 | 49.83 | −0.13 | −0.2 | −0.06 |
| Zanjan | 42.44 | 34.25 | 51.99 | 37.43 | 29.29 | 46.57 | −0.12 | −0.19 | −0.05 |
| All-ages counts estimates | | | | | | | | | |
| Iran | 32,083.13 | 25,531.07 | 40,402.50 | 40,908.57 | 32,066.60 | 50,434.56 | 0.28 | 0.13 | 0.45 |
| Alborz | 852.84 | 677.61 | 1,065.37 | 1,738.63 | 1,360.77 | 2,121.95 | 1.04 | 0.79 | 1.34 |

*(Continued)*

**Table 2.** (*Continued*)

| Location | Year | | | | | | | | |
|---|---|---|---|---|---|---|---|---|---|
| | 1990 | | | 2021 | | | Percentage change 1990–2021 | | |
| | Value | Lower | Upper | Value | Lower | Upper | Value | Lower | Upper |
| Ardebil | 726.2 | 571.52 | 906.22 | 674.68 | 530.38 | 833.61 | −0.07 | −0.2 | 0.08 |
| Bushehr | 428.98 | 339.05 | 541.65 | 618.96 | 485.3 | 763.99 | 0.44 | 0.28 | 0.64 |
| Chahar Mahaal and Bakhtiari | 484.87 | 381.39 | 608.4 | 507.2 | 402.47 | 632.45 | 0.05 | −0.09 | 0.21 |
| East Azarbayejan | 1,905.05 | 1,495.14 | 2,426.58 | 1,990.47 | 1,551.77 | 2,469.60 | 0.04 | −0.1 | 0.21 |
| Fars | 2,468.30 | 1,952.91 | 3,084.60 | 2,866.83 | 2,245.32 | 3,581.91 | 0.16 | 0.02 | 0.34 |
| Gilan | 1,290.10 | 1,030.44 | 1,602.01 | 1,280.72 | 1,020.46 | 1,574.78 | −0.01 | −0.15 | 0.16 |
| Golestan | 962.84 | 761.62 | 1,221.29 | 1,026.12 | 811.59 | 1,266.94 | 0.07 | −0.06 | 0.21 |
| Hamadan | 1,108.85 | 875.84 | 1,399.66 | 988.45 | 775.47 | 1,221.09 | −0.11 | −0.23 | 0.03 |
| Hormozgan | 448.97 | 355.86 | 563.39 | 901.75 | 699.31 | 1,116.00 | 1.01 | 0.76 | 1.24 |
| Ilam | 399.7 | 319.8 | 502.48 | 449.74 | 355.03 | 561.26 | 0.13 | −0.04 | 0.32 |
| Isfahan | 1,645.50 | 1,299.43 | 2,083.79 | 2,271.98 | 1,780.54 | 2,781.57 | 0.38 | 0.2 | 0.6 |
| Kerman | 928.84 | 731.49 | 1,170.48 | 1,594.94 | 1,231.80 | 1,995.61 | 0.72 | 0.5 | 0.97 |
| Kermanshah | 1,634.78 | 1,304.50 | 2,037.65 | 1,315.87 | 1,040.43 | 1,633.22 | −0.2 | −0.3 | −0.07 |
| Khorasan-e-Razavi | 2,429.10 | 1,920.42 | 3,086.05 | 3,057.46 | 2,402.54 | 3,775.67 | 0.26 | 0.1 | 0.43 |
| Khuzestan | 1,843.14 | 1,443.24 | 2,343.91 | 2,517.80 | 1,960.87 | 3,115.10 | 0.37 | 0.21 | 0.55 |
| Kohgiluyeh and Boyer-Ahmad | 357.38 | 282.13 | 458.36 | 506.33 | 392.96 | 638.21 | 0.42 | 0.22 | 0.65 |
| Kurdistan | 853.51 | 672.77 | 1,065.77 | 841.78 | 660.22 | 1,047.65 | −0.01 | −0.15 | 0.14 |
| Lorestan | 1,248.81 | 988.24 | 1,581.47 | 1,189.21 | 934.28 | 1,464.29 | −0.05 | −0.17 | 0.1 |
| Markazi | 612.27 | 485.14 | 771.58 | 601.49 | 474.71 | 749.9 | −0.02 | −0.15 | 0.13 |
| Mazandaran | 1,169.66 | 926.07 | 1,457.30 | 1,577.38 | 1,235.91 | 1,932.35 | 0.35 | 0.18 | 0.58 |
| North Khorasan | 406.36 | 319.96 | 508.8 | 429.77 | 339.25 | 534.98 | 0.06 | −0.06 | 0.2 |
| Qazvin | 361.49 | 283.59 | 457.61 | 580.87 | 457.06 | 719.76 | 0.61 | 0.41 | 0.88 |
| Qom | 277.67 | 219.6 | 354.37 | 566.56 | 437.16 | 706 | 1.04 | 0.77 | 1.32 |
| Semnan | 183.86 | 146.88 | 232.81 | 328.01 | 256.54 | 410.98 | 0.78 | 0.56 | 1.03 |
| Sistan and Baluchistan | 468.26 | 366.97 | 603.82 | 1,248.74 | 958.94 | 1,614.82 | 1.67 | 1.45 | 1.92 |
| South Khorasan | 268.34 | 213.38 | 339.46 | 383.37 | 300.94 | 478.68 | 0.43 | 0.27 | 0.61 |
| Tehran | 4,091.57 | 3,277.09 | 5,103.61 | 5,986.92 | 4,666.01 | 7,380.59 | 0.46 | 0.3 | 0.66 |
| West Azarbayejan | 1,580.07 | 1,241.56 | 1,994.11 | 1,948.27 | 1,544.81 | 2,397.27 | 0.23 | 0.08 | 0.42 |
| Yazd | 291.25 | 229.19 | 363.95 | 488.96 | 380.83 | 613.11 | 0.68 | 0.49 | 0.89 |
| Zanjan | 354.59 | 279.96 | 448.1 | 429.31 | 337.33 | 532.67 | 0.21 | 0.04 | 0.41 |

### DALYs of self-harm in Iran from 1990 to 2021

Age-standardized rate DALYs (per 100,000) of self-harm in 1990 was 350.18 (95% UI: 287.04–382.31) versus 226.63 (95% UI: 202.91–250.46) in 2021; the percentage change from 1990 to 2021 was −0.35% and showed a decreased DALYs. All-ages count estimates increased from 188,787 (95% UI: 148,375–206,949) in 1990 to 203,257 (95% UI: 183,865–225,136) in 2021vs. ; the percentage change from 1990 to 2021 was 0.08 (Table 3 and Figure 3).

### Suicide mortality in Iran from 1990 to 2021

Age-standardized rate (per 100,000) of suicide mortality in 1990 was 6.26 (95% UI: 5.26–6.9) versus 4.12 (95% UI: 3.72–4.58) in 2021; the percentage change from 1990 to 2021 was −0.34% and showed a decreased trend. The incidence of suicide-related mortality in 2021 amounted to 3,708 individuals. This figure indicates an increase when compared to 1990, during which the recorded number of suicide deaths was 3,069 (Table 4 and Figure 4).

### The sex-specific burden of self-harm and suicide mortality in Iran

There are differences between males and females in the prevalence, incidence, DALYs and suicide mortality of self-harm. In 1990, the prevalence of self-harm was lower in males than in females. For males, the age-standardized rate (per 100,000) was 136.66 (95% UI:

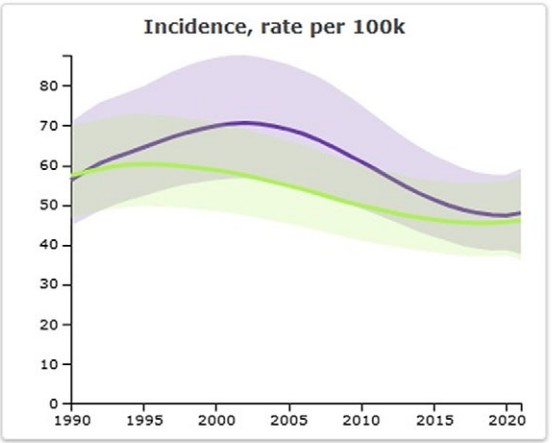

Legend

■ Iran (Islamic Republic of), Both sexes, All ages, Self-harm

■ Iran (Islamic Republic of), Both sexes, Age-standardized, Self-harm

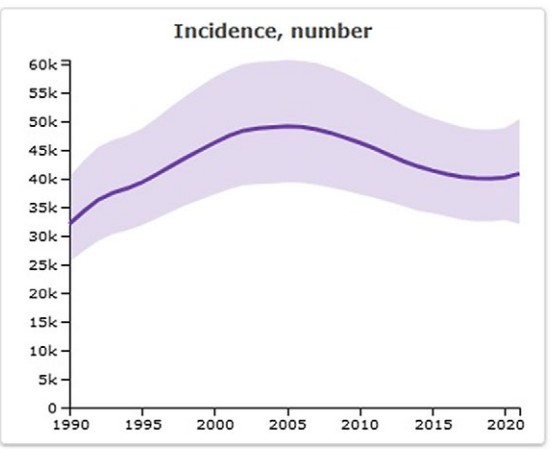

Legend

■ Iran (Islamic Republic of), Both sexes, All ages, Self-harm

**Figure 2.** Trend in incidence of self-harm in Iran, 1990–2021.

**Table 3.** All-ages counts and age-standardized rate (per 100,000) DALYs of self-harm in Iran, stratified by provinces, 1990–2021

| | Year | | | | | | | | |
| --- | --- | --- | --- | --- | --- | --- | --- | --- | --- |
| | 1990 | | | 2021 | | | Percentage change 1990–2021 | | |
| Location | Value | Lower | Upper | Value | Lower | Upper | Value | Lower | Upper |
| Age-standardized rate DALYs (per 100,000) | | | | | | | | | |
| Iran | 350.18 | 287.04 | 382.31 | 226.63 | 202.91 | 250.46 | −0.35 | −0.42 | −0.22 |
| Alborz | 493.12 | 252.38 | 631.4 | 356.58 | 190.9 | 454.22 | −0.28 | −0.47 | 0.11 |
| Ardebil | 408.97 | 295.12 | 508.74 | 291.96 | 204.31 | 342.93 | −0.29 | −0.44 | −0.01 |
| Bushehr | 341.13 | 239.41 | 431.86 | 171.32 | 142.1 | 204.44 | −0.5 | −0.62 | −0.27 |
| Chahar Mahaal and Bakhtiari | 401.81 | 259.98 | 502.95 | 184.18 | 140.85 | 232.39 | −0.54 | −0.66 | −0.33 |
| East Azarbayejan | 415.46 | 313.52 | 513.48 | 266.1 | 211.07 | 324.22 | −0.36 | −0.51 | −0.14 |
| Fars | 406.88 | 311.49 | 501.78 | 319.17 | 254.46 | 393.1 | −0.22 | −0.4 | 0.05 |

*(Continued)*

**Table 3.** (*Continued*)

| | Year | | | | | | | | |
|---|---|---|---|---|---|---|---|---|---|
| | 1990 | | | 2021 | | | Percentage change 1990–2021 | | |
| Location | Value | Lower | Upper | Value | Lower | Upper | Value | Lower | Upper |
| Gilan | 361.58 | 276.02 | 451.34 | 244.75 | 193.76 | 299.89 | −0.32 | −0.49 | −0.11 |
| Golestan | 489.57 | 300.89 | 616.74 | 309.1 | 243.07 | 366.57 | −0.37 | −0.51 | −0.13 |
| Hamadan | 555.83 | 344.06 | 691.64 | 416.68 | 255.41 | 513.88 | −0.25 | −0.44 | −0.01 |
| Hormozgan | 356.45 | 275.81 | 454.42 | 260.18 | 207.84 | 320.75 | −0.27 | −0.47 | 0 |
| Ilam | 880.89 | 320.57 | 1,132.86 | 566.95 | 253.28 | 691.14 | −0.36 | −0.51 | −0.08 |
| Isfahan | 178.11 | 139.92 | 268.33 | 144.55 | 114.9 | 198.61 | −0.19 | −0.4 | 0.1 |
| Kerman | 377.8 | 301.1 | 471.96 | 233.57 | 192.98 | 291.97 | −0.38 | −0.54 | −0.16 |
| Kermanshah | 785.01 | 440.5 | 977.94 | 477.91 | 271.32 | 582.46 | −0.39 | −0.54 | −0.18 |
| Khorasan-e-Razavi | 355.79 | 273.46 | 438.88 | 198.85 | 161.85 | 242.65 | −0.44 | −0.57 | −0.24 |
| Khuzestan | 346.58 | 276.61 | 431.72 | 253.28 | 212.23 | 299.43 | −0.27 | −0.44 | −0.04 |
| Kohgiluyeh and Boyer-Ahmad | 573.56 | 335.88 | 710.31 | 434.18 | 298.98 | 520.95 | −0.24 | −0.41 | 0 |
| Kurdistan | 421.82 | 305.53 | 534.14 | 225.02 | 181.42 | 270.28 | −0.47 | −0.6 | −0.25 |
| Lorestan | 724.79 | 305.66 | 933.62 | 377.43 | 192.99 | 464.12 | −0.48 | −0.62 | −0.25 |
| Markazi | 294.71 | 231.32 | 379.33 | 175.1 | 141.12 | 228.91 | −0.41 | −0.56 | −0.19 |
| Mazandaran | 198.17 | 155.78 | 260.8 | 158.73 | 129.76 | 217.98 | −0.2 | −0.39 | 0.04 |
| North Khorasan | 374.07 | 282.56 | 457.11 | 238.5 | 200.23 | 280.85 | −0.36 | −0.51 | −0.12 |
| Qazvin | 223.39 | 175.79 | 292.66 | 149.51 | 119.74 | 189.36 | −0.33 | −0.51 | −0.1 |
| Qom | 231.49 | 178.3 | 335.87 | 179.69 | 144.44 | 231.96 | −0.22 | −0.46 | 0.09 |
| Semnan | 215.17 | 171.28 | 280.5 | 121.24 | 94.55 | 163.1 | −0.44 | −0.59 | −0.25 |
| Sistan and Baluchistan | 190 | 142.26 | 283.89 | 198.58 | 155.68 | 330.4 | 0.05 | −0.22 | 0.45 |
| South Khorasan | 223.24 | 174.56 | 288 | 158.68 | 133.85 | 196.89 | −0.29 | −0.45 | −0.05 |
| Tehran | 170.45 | 130.14 | 232.72 | 106.46 | 83.78 | 153.4 | −0.38 | −0.53 | −0.1 |
| West Azarbayejan | 539.35 | 371.79 | 675.15 | 334.31 | 234.64 | 396.99 | −0.38 | −0.52 | −0.18 |
| Yazd | 200.33 | 160.86 | 283.47 | 120.79 | 95.01 | 167.71 | −0.4 | −0.56 | −0.18 |
| Zanjan | 233.32 | 183.92 | 287.05 | 130.58 | 106.69 | 169.15 | −0.44 | −0.57 | −0.19 |
| All-ages counts estimates | | | | | | | | | |
| Iran | 188,787.36 | 148,375.22 | 206,949.05 | 203,257.54 | 183,865.89 | 225,136.26 | 0.08 | −0.04 | 0.34 |
| Alborz | 6,687.32 | 3,383.84 | 8,655.34 | 11,694.80 | 6,202.08 | 14,982.35 | 0.75 | 0.29 | 1.66 |
| Ardebil | 4,298.47 | 3,001.06 | 5,401.70 | 4,095.63 | 2,884.61 | 4,772.64 | −0.05 | −0.25 | 0.35 |
| Bushehr | 2,226.34 | 1,495.73 | 2,824.44 | 2,342.69 | 1,941.82 | 2,791.26 | 0.05 | −0.2 | 0.56 |
| Chahar Mahaal and Bakhtiari | 2,572.76 | 1,586.08 | 3,295.07 | 1,958.68 | 1,497.03 | 2,464.93 | −0.24 | −0.44 | 0.14 |
| East Azarbayejan | 13,840.83 | 10,046.49 | 17,409.55 | 11,301.21 | 9,024.35 | 13,630.56 | −0.18 | −0.38 | 0.12 |
| Fars | 13,854.68 | 10,114.14 | 17,323.50 | 17,182.30 | 13,656.97 | 21,111.36 | 0.24 | −0.07 | 0.69 |
| Gilan | 7,986.56 | 5,896.88 | 10,049.17 | 6,744.62 | 5,337.00 | 8,212.65 | −0.16 | −0.35 | 0.12 |
| Golestan | 6,024.01 | 3,505.53 | 7,683.93 | 6,186.09 | 4,854.80 | 7,301.54 | 0.03 | −0.21 | 0.5 |
| Hamadan | 8,840.85 | 5,310.55 | 11,115.27 | 7,578.78 | 4,638.73 | 9,382.85 | −0.14 | −0.36 | 0.14 |
| Hormozgan | 2,869.23 | 2,165.39 | 3,655.39 | 5,477.61 | 4,365.06 | 6,742.94 | 0.91 | 0.36 | 1.69 |
| Ilam | 3,497.29 | 1,202.58 | 4,504.89 | 3,808.27 | 1,716.84 | 4,634.00 | 0.09 | −0.18 | 0.64 |
| Isfahan | 6,601.83 | 5,114.99 | 9,719.58 | 8,094.64 | 6,453.58 | 11,114.41 | 0.23 | −0.11 | 0.67 |
| Kerman | 6,300.35 | 4,955.29 | 7,911.67 | 8,304.06 | 6,854.86 | 10,420.84 | 0.32 | −0.04 | 0.78 |
| Kermanshah | 12,517.38 | 6,747.37 | 15,817.70 | 10,157.17 | 5,823.33 | 12,417.83 | −0.19 | −0.4 | 0.11 |
| Khorasan-e-Razavi | 15,211.79 | 11,724.65 | 18,831.81 | 14,243.37 | 11,589.26 | 17,306.93 | −0.06 | −0.28 | 0.31 |

(*Continued*)

**Table 3.** (*Continued*)

| Location | Year | | | | | | | | |
|---|---|---|---|---|---|---|---|---|---|
| | 1990 | | | 2021 | | | Percentage change 1990–2021 | | |
| | Value | Lower | Upper | Value | Lower | Upper | Value | Lower | Upper |
| Khuzestan | 9,906.02 | 7,774.19 | 12,392.73 | 12,949.45 | 10,892.92 | 15,186.51 | 0.31 | −0.01 | 0.73 |
| Kohgiluyeh and Boyer-Ahmad | 2,701.46 | 1,511.41 | 3,379.37 | 3,666.89 | 2,593.28 | 4,395.43 | 0.36 | 0.04 | 0.89 |
| Kurdistan | 4,885.55 | 3,329.27 | 6,228.69 | 4,189.00 | 3,395.40 | 4,981.09 | −0.14 | −0.36 | 0.22 |
| Lorestan | 10,132.03 | 3,964.18 | 13,262.95 | 7,291.35 | 3,693.00 | 9,030.03 | −0.28 | −0.47 | 0.1 |
| Markazi | 3,375.52 | 2,647.18 | 4,259.36 | 2,716.78 | 2,183.41 | 3,555.95 | −0.2 | −0.41 | 0.1 |
| Mazandaran | 4,995.86 | 3,910.74 | 6,458.09 | 5,829.47 | 4,793.05 | 8,074.73 | 0.17 | −0.11 | 0.52 |
| North Khorasan | 2,228.20 | 1,673.37 | 2,742.59 | 2,124.05 | 1,784.06 | 2,507.51 | −0.05 | −0.29 | 0.31 |
| Qazvin | 1,920.77 | 1,505.80 | 2,447.16 | 2,193.83 | 1,767.28 | 2,804.78 | 0.14 | −0.16 | 0.55 |
| Qom | 1,565.97 | 1,188.84 | 2,266.52 | 2,768.63 | 2,235.96 | 3,585.56 | 0.77 | 0.24 | 1.5 |
| Semnan | 958.21 | 762.6 | 1,229.19 | 1,061.99 | 835.54 | 1,432.08 | 0.11 | −0.19 | 0.49 |
| Sistan and Baluchistan | 2,329.67 | 1,674.97 | 3,142.36 | 6,234.16 | 4,879.94 | 10,396.56 | 1.68 | 0.96 | 2.94 |
| South Khorasan | 1,297.16 | 1,005.52 | 1,660.15 | 1,452.93 | 1,223.74 | 1,797.36 | 0.12 | −0.13 | 0.52 |
| Tehran | 13,837.11 | 10,528.42 | 18,855.65 | 16,561.57 | 13,082.46 | 23,630.01 | 0.2 | −0.11 | 0.73 |
| West Azarbayejan | 12,249.07 | 8,154.51 | 15,321.40 | 12,016.01 | 8,433.89 | 14,271.46 | −0.02 | −0.24 | 0.33 |
| Yazd | 1,230.79 | 976 | 1,682.76 | 1,492.94 | 1,178.11 | 2,081.85 | 0.21 | −0.11 | 0.66 |
| Zanjan | 1,844.30 | 1,411.32 | 2,281.83 | 1,538.57 | 1,257.01 | 1,997.43 | −0.17 | −0.37 | 0.24 |

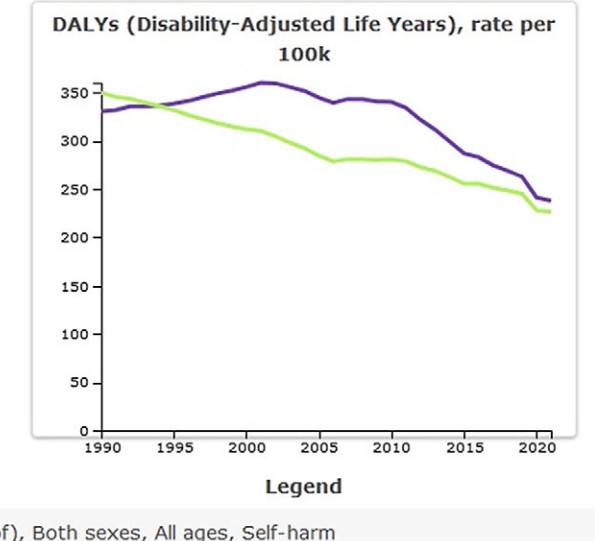

**Figure 3.** Trend in DALYs of self-harm in Iran, 1990–2021.

116.78–162.01); for females, it was 212.79 (95% UI: 176.25–258.02). In 2021, the prevalence of self-harm was higher in males (137.62 per 100,000) than in females (124.82 per 100,000). Since 2015, a notable shift in the prevalence rates of self-harm among men and women has been observed, signaling an epidemiological transformation. By 2021, the incidence of self-harm reached a total of 64,913 cases among males (95% UI: 54,867–76,963) and 57,867 cases among females (95% UI: 48,020–70,468) (Table 5 and Figure 5).

The incidence of self-harm in males and females has had a different trend. Since 1990, the incidence of self-harm in males has shown a relatively stable trend with a slight increase. In contrast, a decreasing trend can be seen in females; however, in 2020, the incidence was close to that of males, and in 2021, it shows an increasing trend again (Table 5 and Figure 6). In 2021, age-standardized rate of suicide mortality (per 100,000) in males was 6.04 (95% UI: 5.14–6.67), and in females was 2.14 (95% UI: 1.85–2.74).

**Table 4.** All-ages counts and age-standardized rate (per 100,000) of suicide mortality in Iran, stratified by provinces, 1990–2021

| | Year | | | | | | | | |
| --- | --- | --- | --- | --- | --- | --- | --- | --- | --- |
| | 1990 | | | 2021 | | | Percentage change 1990–2021 | | |
| Location | Value | Lower | Upper | Value | Lower | Upper | Value | Lower | Upper |
| Age-standardized rate suicide mortality (per 100,000) | | | | | | | | | |
| Iran | 6.26 | 5.26 | 6.9 | 4.12 | 3.72 | 4.58 | −0.34 | −0.41 | −0.19 |
| Alborz | 9.26 | 4.67 | 11.95 | 6.64 | 3.5 | 8.49 | −0.28 | −0.47 | 0.1 |
| Ardebil | 7.3 | 5.28 | 9.1 | 5.52 | 3.86 | 6.46 | −0.24 | −0.41 | 0.04 |
| Bushehr | 6.15 | 4.36 | 7.74 | 3.29 | 2.74 | 3.89 | −0.46 | −0.59 | −0.23 |
| Chahar Mahaal and Bakhtiari | 7.28 | 4.78 | 9.08 | 3.49 | 2.65 | 4.43 | −0.52 | −0.64 | −0.28 |
| East Azarbayejan | 7.38 | 5.65 | 9.13 | 4.89 | 3.88 | 5.92 | −0.34 | −0.49 | −0.12 |
| Fars | 7.21 | 5.61 | 8.9 | 5.62 | 4.44 | 6.88 | −0.22 | −0.41 | 0.06 |
| Gilan | 6.71 | 5.22 | 8.29 | 4.56 | 3.58 | 5.56 | −0.32 | −0.49 | −0.11 |
| Golestan | 8.59 | 5.36 | 10.68 | 5.67 | 4.44 | 6.69 | −0.34 | −0.49 | −0.1 |
| Hamadan | 9.97 | 6.17 | 12.47 | 7.56 | 4.54 | 9.39 | −0.24 | −0.43 | 0.03 |
| Hormozgan | 6.47 | 4.95 | 8.2 | 4.65 | 3.73 | 5.75 | −0.28 | −0.48 | −0.02 |
| Ilam | 16.24 | 5.73 | 20.62 | 10.68 | 4.58 | 12.93 | −0.34 | −0.49 | −0.07 |
| Isfahan | 3.12 | 2.4 | 4.86 | 2.57 | 2.03 | 3.59 | −0.18 | −0.4 | 0.13 |
| Kerman | 6.75 | 5.4 | 8.45 | 4.2 | 3.44 | 5.35 | −0.38 | −0.54 | −0.15 |
| Kermanshah | 14.09 | 7.97 | 17.46 | 8.64 | 4.81 | 10.57 | −0.39 | −0.54 | −0.16 |
| Khorasan-e-Razavi | 6.4 | 5.06 | 7.95 | 3.67 | 2.98 | 4.46 | −0.43 | −0.56 | −0.23 |
| Khuzestan | 6.14 | 4.88 | 7.63 | 4.53 | 3.82 | 5.35 | −0.26 | −0.43 | −0.03 |
| Kohgiluyeh and Boyer-Ahmad | 9.71 | 5.77 | 12.05 | 7.41 | 5.11 | 8.88 | −0.24 | −0.4 | 0.01 |
| Kurdistan | 7.42 | 5.4 | 9.47 | 4.09 | 3.3 | 4.94 | −0.45 | −0.59 | −0.22 |
| Lorestan | 13.2 | 5.59 | 16.99 | 7.28 | 3.57 | 8.93 | −0.45 | −0.58 | −0.22 |
| Markazi | 5.23 | 4.14 | 6.98 | 3.14 | 2.51 | 4.19 | −0.4 | −0.56 | −0.18 |
| Mazandaran | 3.5 | 2.74 | 4.74 | 2.86 | 2.33 | 3.96 | −0.18 | −0.37 | 0.08 |
| North Khorasan | 6.52 | 4.87 | 7.99 | 4.28 | 3.61 | 5.04 | −0.34 | −0.5 | −0.07 |
| Qazvin | 4.06 | 3.22 | 5.58 | 2.83 | 2.27 | 3.63 | −0.3 | −0.48 | −0.06 |
| Qom | 4.27 | 3.27 | 6.3 | 3.28 | 2.62 | 4.23 | −0.23 | −0.47 | 0.07 |
| Semnan | 3.95 | 3.14 | 5.32 | 2.27 | 1.79 | 3.09 | −0.43 | −0.58 | −0.23 |
| Sistan and Baluchistan | 3.4 | 2.55 | 5.67 | 3.47 | 2.72 | 5.93 | 0.02 | −0.23 | 0.42 |
| South Khorasan | 3.98 | 3.12 | 5.25 | 2.88 | 2.42 | 3.64 | −0.28 | −0.44 | −0.02 |
| Tehran | 3.11 | 2.34 | 4.33 | 1.97 | 1.55 | 2.87 | −0.37 | −0.53 | −0.07 |
| West Azarbayejan | 9.5 | 6.74 | 11.83 | 6.06 | 4.22 | 7.2 | −0.36 | −0.5 | −0.15 |
| Yazd | 3.63 | 2.92 | 5.36 | 2.19 | 1.74 | 3.11 | −0.4 | −0.56 | −0.17 |
| Zanjan | 4.23 | 3.39 | 5.23 | 2.46 | 2.01 | 3.22 | −0.42 | −0.55 | −0.17 |
| All-ages counts estimates | | | | | | | | | |
| Iran (Islamic Republic of) | 3,069.66 | 2,463.49 | 3,362.63 | 3,708.56 | 3,354.15 | 4,117.61 | 0.21 | 0.07 | 0.5 |
| Alborz | 112.25 | 56.14 | 145.25 | 217.09 | 112.52 | 278.66 | 0.93 | 0.43 | 2 |
| Ardebil | 69.12 | 48.96 | 86.94 | 77.61 | 54.53 | 90.84 | 0.12 | −0.11 | 0.58 |
| Bushehr | 35.93 | 24.4 | 45.81 | 42.91 | 35.72 | 50.92 | 0.19 | −0.1 | 0.79 |
| Chahar Mahaal and Bakhtiari | 41.83 | 25.94 | 53.2 | 36.79 | 28.31 | 46.41 | −0.12 | −0.35 | 0.32 |
| East Azarbayejan | 222.88 | 163.9 | 278.89 | 210.69 | 168.22 | 256.38 | −0.05 | −0.29 | 0.29 |
| Fars | 223.18 | 166.22 | 277.95 | 305.49 | 243.09 | 375.63 | 0.37 | 0.02 | 0.86 |

(Continued)

**Table 4.** (*Continued*)

| | Year | | | | | | | | |
|---|---|---|---|---|---|---|---|---|---|
| | 1990 | | | 2021 | | | Percentage change 1990–2021 | | |
| Location | Value | Lower | Upper | Value | Lower | Upper | Value | Lower | Upper |
| Gilan | 135.4 | 101.72 | 169.4 | 131.37 | 103.42 | 160.69 | −0.03 | −0.26 | 0.28 |
| Golestan | 96.3 | 56.94 | 121.95 | 112.94 | 89.32 | 134.02 | 0.17 | −0.09 | 0.67 |
| Hamadan | 144.66 | 87.42 | 180.94 | 141.58 | 84.61 | 177.2 | −0.02 | −0.27 | 0.31 |
| Hormozgan | 47.24 | 35.7 | 61.18 | 94.83 | 75.59 | 117 | 1.01 | 0.45 | 1.8 |
| Ilam | 57.15 | 19.34 | 73.91 | 70.78 | 30.48 | 86.22 | 0.24 | −0.06 | 0.83 |
| Isfahan | 105.42 | 80.84 | 159.8 | 147.13 | 116.7 | 204.98 | 0.4 | 0.01 | 0.93 |
| Kerman | 102.64 | 80.31 | 128.48 | 146.82 | 120.13 | 186.11 | 0.43 | 0.05 | 0.94 |
| Kermanshah | 203.33 | 109.99 | 256.36 | 186.67 | 104.82 | 228.98 | −0.08 | −0.31 | 0.26 |
| Khorasan-e-Razavi | 247.99 | 190.24 | 307.48 | 259.84 | 209.17 | 314.53 | 0.05 | −0.21 | 0.44 |
| Khuzestan | 159.37 | 126.21 | 199.72 | 228.89 | 192.68 | 270.19 | 0.44 | 0.09 | 0.9 |
| Kohgiluyeh and Boyer-Ahmad | 41.58 | 23.27 | 52.13 | 62.25 | 43.42 | 74.27 | 0.5 | 0.15 | 1.08 |
| Kurdistan | 78.09 | 54.4 | 99.75 | 76.24 | 61.66 | 90.81 | −0.02 | −0.26 | 0.39 |
| Lorestan | 165.74 | 66.09 | 216.64 | 139.13 | 67.9 | 171.91 | −0.16 | −0.39 | 0.26 |
| Markazi | 54.87 | 42.7 | 70.94 | 49.92 | 39.55 | 66.64 | −0.09 | −0.33 | 0.26 |
| Mazandaran | 80.41 | 62.45 | 106.15 | 108.66 | 88.26 | 152.86 | 0.35 | 0.03 | 0.79 |
| North Khorasan | 35.51 | 26.21 | 43.78 | 37.92 | 31.95 | 44.84 | 0.07 | −0.21 | 0.49 |
| Qazvin | 31.46 | 24.44 | 40.78 | 41.29 | 33.04 | 53.25 | 0.31 | −0.02 | 0.8 |
| Qom | 25.75 | 19.64 | 37.82 | 49.67 | 39.95 | 64.19 | 0.93 | 0.34 | 1.75 |
| Semnan | 16.21 | 12.86 | 21.11 | 19.59 | 15.53 | 26.77 | 0.21 | −0.12 | 0.63 |
| Sistan and Baluchistan | 37.63 | 27.62 | 54.31 | 103.54 | 80.93 | 176.23 | 1.75 | 1.02 | 2.96 |
| South Khorasan | 21.33 | 16.44 | 27.63 | 26.09 | 21.8 | 32.83 | 0.22 | −0.07 | 0.65 |
| Tehran | 231.21 | 173.85 | 319.4 | 312.09 | 243.11 | 451.58 | 0.35 | −0.01 | 1 |
| West Azarbayejan | 194.55 | 131.62 | 243.98 | 214.92 | 149.49 | 257.61 | 0.1 | −0.14 | 0.49 |
| Yazd | 20.4 | 16.21 | 28.9 | 26.7 | 21.14 | 37.83 | 0.31 | −0.05 | 0.81 |
| Zanjan | 30.27 | 23.66 | 37.37 | 29.13 | 23.64 | 38.12 | −0.04 | −0.27 | 0.39 |

The suicide mortality rate in males compared to females has been higher since 1990 and has increased over time. In 2021, there were 2,398 suicide mortality among males, compared with 906 suicide mortality among females (Table 6 and Figure 7).

### The age-specific burden of self-harm and suicide mortality in Iran

In 2021, the age-specific rate of suicide mortality (per 100,000) was highest among individuals aged 15–39 years, at 6.82 (95% UI: 6.09–7.48). For both males and females, the highest age-specific rate of suicide mortality (per 100,000) was among individuals aged 15–39 years, at 10.01 (95% UI: 8.57–11.06) and 3.51 (95% UI: 3–4.42), respectively (Table 7 and Figure 8).

### The burden of self-harm and suicide mortality stratified by provinces

The subnational burden of self-harm and suicide mortality showed that the highest age-standardized prevalence rate per 100,000 in 2021 was in Ilam (188.33 [95% UI: 157.11–223.85]), and the lowest was in Zanjan (106.84 [95% UI: 90.84–127.12]) (Table 1 and Figure 9). The highest self-harm counts were in Tehran (18,736 [95% UI: 15,979–22,234]), and the lowest was in Semnan (945 [95% UI: 799–1,128]) (Table 1 and Figure 10). The highest age-standardized suicide mortality rate per 100,000 was in Ilam (10.68 [95% UI: 4.58–12.93]) (Table 4 and Figure 11).

### Discussion

The present study aimed to investigate the impact of self-harm and suicide mortality across different sex and age groups in Iran from 1990 to 2021. It additionally sought to estimate the prevalence, incidence, disability and mortality rates associated with these issues, analyzing data at both national and subnational levels to provide a comprehensive understanding of their distribution and burden.

Over the past three decades, the patterns of self-harm and suicide mortality in Iran have exhibited a decline. The prevalence, incidence, disability attributed to self-harm and the rates of suicide mortality have shown noticeable reductions compared to figures from 1990. The conclusion of the war against Iran in 1988 marked a period when mental health disorders escalated due to the

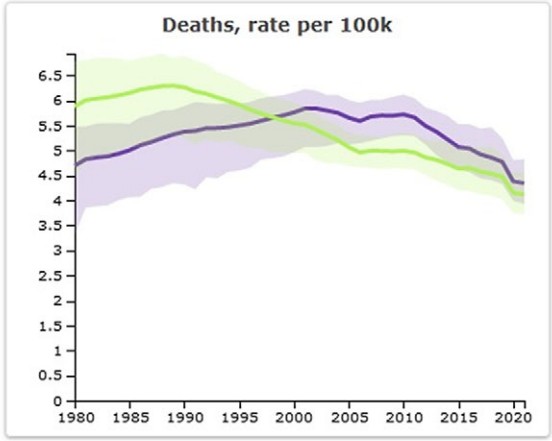

### Legend

- Iran (Islamic Republic of), Both sexes, All ages, Self-harm
- Iran (Islamic Republic of), Both sexes, Age-standardized, Self-harm

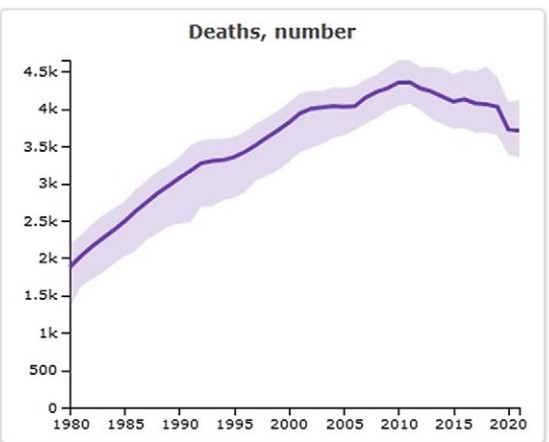

### Legend

- Iran (Islamic Republic of), Both sexes, All ages, Self-harm
- Iran (Islamic Republic of), Both sexes, Age-standardized, Self-harm

**Figure 4.** Trend in suicide mortality in Iran, 1990–2021.

**Table 5.** Sex-specific prevalence, incidence, and DALYs of self-harm in Iran, stratified by provinces, 1990–2021

| | | Year | | | | | | | | |
| | | 1990 | | | 2021 | | | Percentage change 1990–2021 | | |
| Location | Sex | Value | Lower | Upper | Value | Lower | Upper | Value | Lower | Upper |
|---|---|---|---|---|---|---|---|---|---|---|
| Age-standardized rate prevalence (per 100,000) | | | | | | | | | | |
| Iran (Islamic Republic of) | Males | 136.66 | 116.78 | 162.01 | 137.62 | 116.74 | 162.53 | 0.01 | −0.03 | 0.04 |
| Alborz | Males | 163.83 | 138.08 | 193.6 | 170.83 | 143.83 | 203.19 | 0.04 | 0.01 | 0.08 |
| Ardebil | Males | 147.11 | 125.39 | 174.27 | 146.43 | 123.87 | 173.2 | 0 | −0.05 | 0.04 |
| Bushehr | Males | 108.32 | 93.34 | 127.56 | 115.08 | 99.08 | 134.96 | 0.06 | 0.03 | 0.1 |
| Chahar Mahaal and Bakhtiari | Males | 141.49 | 121.28 | 167.54 | 122.41 | 104.08 | 144.75 | −0.13 | −0.17 | −0.11 |

(*Continued*)

**Table 5.** (*Continued*)

| Location | Sex | Year | | | | | | | | |
| | | 1990 | | | 2021 | | | Percentage change 1990–2021 | | |
| | | Value | Lower | Upper | Value | Lower | Upper | Value | Lower | Upper |
|---|---|---|---|---|---|---|---|---|---|---|
| East Azarbayejan | Males | 146.14 | 122.95 | 175.12 | 150.01 | 126.79 | 177.18 | 0.03 | −0.04 | 0.09 |
| Fars | Males | 156.62 | 132.69 | 186.4 | 154.82 | 130.19 | 182.87 | −0.01 | −0.06 | 0.03 |
| Gilan | Males | 129.48 | 109.09 | 154.08 | 142.85 | 121.01 | 168.34 | 0.1 | 0.05 | 0.16 |
| Golestan | Males | 132.94 | 112.99 | 157.58 | 138.12 | 116.27 | 163.85 | 0.04 | 0 | 0.08 |
| Hamadan | Males | 183.67 | 154.66 | 217.44 | 178.33 | 150.46 | 210.48 | −0.03 | −0.09 | 0.02 |
| Hormozgan | Males | 136.49 | 116.77 | 160.3 | 143.66 | 120.47 | 170.41 | 0.05 | 0.02 | 0.09 |
| Ilam | Males | 174.94 | 147.95 | 207.73 | 185.15 | 155.1 | 219.47 | 0.06 | 0.01 | 0.11 |
| Isfahan | Males | 120.93 | 104.21 | 142.67 | 131.21 | 110.32 | 156.01 | 0.08 | 0.05 | 0.13 |
| Kerman | Males | 130.06 | 110.03 | 155.79 | 135.89 | 114.57 | 161.69 | 0.04 | 0 | 0.09 |
| Kermanshah | Males | 213.47 | 180.1 | 252.98 | 171.91 | 144.46 | 202.43 | −0.19 | −0.24 | −0.16 |
| Khorasan-e-Razavi | Males | 120.01 | 101.73 | 142.93 | 131.06 | 111.92 | 154.32 | 0.09 | 0.04 | 0.15 |
| Khuzestan | Males | 125.51 | 107.76 | 147.84 | 135.26 | 114.71 | 160.87 | 0.08 | 0.04 | 0.12 |
| Kohgiluyeh and Boyer-Ahmad | Males | 152.92 | 128.2 | 185.68 | 161.33 | 133.97 | 191.2 | 0.05 | 0 | 0.11 |
| Kurdistan | Males | 144.67 | 124.12 | 171.59 | 133.17 | 112.12 | 157.88 | −0.08 | −0.12 | −0.04 |
| Lorestan | Males | 170.14 | 144.18 | 200.88 | 162.63 | 136.65 | 191.88 | −0.04 | −0.08 | −0.01 |
| Markazi | Males | 153.06 | 129.43 | 183.8 | 131.17 | 110.5 | 154.49 | −0.14 | −0.19 | −0.1 |
| Mazandaran | Males | 106.78 | 91.57 | 125.97 | 129.96 | 110.33 | 154.04 | 0.22 | 0.17 | 0.26 |
| North Khorasan | Males | 123.89 | 106.34 | 147.15 | 125.66 | 107.34 | 149.21 | 0.01 | −0.03 | 0.06 |
| Qazvin | Males | 94.46 | 81.32 | 111.76 | 127.18 | 107.98 | 149.52 | 0.35 | 0.3 | 0.41 |
| Qom | Males | 129.43 | 110.25 | 153.63 | 130.34 | 109.48 | 155.05 | 0.01 | −0.04 | 0.06 |
| Semnan | Males | 108.7 | 93.72 | 127.54 | 123.29 | 104.92 | 144.83 | 0.13 | 0.1 | 0.17 |
| Sistan and Baluchistan | Males | 98.52 | 85.6 | 115.04 | 127.44 | 108.69 | 150.3 | 0.29 | 0.25 | 0.35 |
| South Khorasan | Males | 99.77 | 85.85 | 116.51 | 124.17 | 106.32 | 146.05 | 0.24 | 0.19 | 0.3 |
| Tehran | Males | 140.72 | 121.56 | 164.86 | 120.41 | 103.19 | 141.56 | −0.14 | −0.17 | −0.12 |
| West Azarbayejan | Males | 136.83 | 114.97 | 165.12 | 152.28 | 128.35 | 178.97 | 0.11 | 0.05 | 0.18 |
| Yazd | Males | 106.22 | 91.71 | 125.49 | 118.59 | 100.71 | 140.09 | 0.12 | 0.08 | 0.17 |
| Zanjan | Males | 94.22 | 81.02 | 111.67 | 112.97 | 96.35 | 132.83 | 0.2 | 0.16 | 0.25 |
| Iran (Islamic Republic of) | Females | 212.79 | 176.25 | 258.02 | 124.82 | 104.03 | 151.66 | −0.41 | −0.44 | −0.39 |
| Alborz | Females | 194.62 | 160.52 | 237.67 | 133.38 | 109.69 | 162.29 | −0.31 | −0.34 | −0.29 |
| Ardebil | Females | 262.38 | 218.03 | 320.49 | 132.92 | 111.12 | 159.89 | −0.49 | −0.53 | −0.46 |
| Bushehr | Females | 263.56 | 218.14 | 319.91 | 137.95 | 115.98 | 164.82 | −0.48 | −0.51 | −0.44 |
| Chahar Mahaal and Bakhtiari | Females | 287.49 | 237.05 | 346.59 | 142.37 | 117.99 | 172.09 | −0.5 | −0.54 | −0.48 |
| East Azarbayejan | Females | 206.17 | 168.84 | 255.19 | 125.54 | 104.22 | 153.25 | −0.39 | −0.42 | −0.36 |
| Fars | Females | 268.48 | 220.89 | 329.47 | 146.49 | 121.12 | 178.87 | −0.45 | −0.48 | −0.43 |
| Gilan | Females | 187.13 | 153.98 | 227.69 | 125.22 | 103.71 | 152.34 | −0.33 | −0.36 | −0.3 |
| Golestan | Females | 325.86 | 267.47 | 409.23 | 157.58 | 131.42 | 190.88 | −0.52 | −0.55 | −0.48 |
| Hamadan | Females | 235.8 | 193.77 | 293.79 | 142.45 | 117.46 | 173.31 | −0.4 | −0.43 | −0.37 |
| Hormozgan | Females | 196.78 | 163.41 | 237.7 | 108.38 | 91.43 | 131.41 | −0.45 | −0.48 | −0.42 |
| Ilam | Females | 391.63 | 322.86 | 477.01 | 191.76 | 159.36 | 229.14 | −0.51 | −0.54 | −0.48 |
| Isfahan | Females | 147.17 | 122.35 | 179.41 | 102.96 | 86.62 | 124.11 | −0.3 | −0.34 | −0.27 |
| Kerman | Females | 201.38 | 166.13 | 245.48 | 126.81 | 105.55 | 155.48 | −0.37 | −0.4 | −0.34 |

(*Continued*)

**Table 5.** (*Continued*)

| Location | Sex | Year | | | | | | | | |
|---|---|---|---|---|---|---|---|---|---|---|
| | | 1990 | | | 2021 | | | Percentage change 1990–2021 | | |
| | | Value | Lower | Upper | Value | Lower | Upper | Value | Lower | Upper |
| Kermanshah | Females | 396.03 | 327.12 | 494.24 | 186.02 | 152.78 | 228.87 | −0.53 | −0.56 | −0.5 |
| Khorasan-e-Razavi | Females | 219.32 | 181.16 | 269.09 | 119.19 | 99.32 | 143.53 | −0.46 | −0.49 | −0.42 |
| Khuzestan | Females | 247.81 | 204.57 | 301.81 | 145.62 | 119.92 | 177.73 | −0.41 | −0.44 | −0.38 |
| Kohgiluyeh and Boyer-Ahmad | Females | 307.75 | 252.04 | 385.9 | 177.37 | 145.18 | 217.53 | −0.42 | −0.45 | −0.39 |
| Kurdistan | Females | 295.53 | 243.08 | 367.49 | 129.3 | 107.95 | 156.02 | −0.56 | −0.59 | −0.53 |
| Lorestan | Females | 339.58 | 281.1 | 417.86 | 185.75 | 153.94 | 224.89 | −0.45 | −0.48 | −0.42 |
| Markazi | Females | 172.34 | 143.2 | 209.3 | 97.89 | 82.65 | 117.82 | −0.43 | −0.47 | −0.4 |
| Mazandaran | Females | 161.91 | 133.47 | 198.03 | 114.71 | 94.21 | 139.21 | −0.29 | −0.32 | −0.26 |
| North Khorasan | Females | 288.87 | 239.4 | 358.38 | 154.63 | 128.55 | 187.04 | −0.46 | −0.49 | −0.44 |
| Qazvin | Females | 151.48 | 126.66 | 182.54 | 105.33 | 88.65 | 127.22 | −0.3 | −0.33 | −0.28 |
| Qom | Females | 117.62 | 99.64 | 140.8 | 85.98 | 73.07 | 103.62 | −0.27 | −0.3 | −0.24 |
| Semnan | Females | 137.72 | 114.97 | 167.14 | 95.06 | 80.27 | 114.2 | −0.31 | −0.34 | −0.28 |
| Sistan and Baluchistan | Females | 134.15 | 112.85 | 160.44 | 115.94 | 97.34 | 139.55 | −0.14 | −0.16 | −0.11 |
| South Khorasan | Females | 178.25 | 148.6 | 216.38 | 120.68 | 101.08 | 147.06 | −0.32 | −0.35 | −0.3 |
| Tehran | Females | 153.7 | 129.42 | 183.73 | 98.65 | 83.33 | 118.84 | −0.36 | −0.39 | −0.33 |
| West Azarbayejan | Females | 285.56 | 234.25 | 358.33 | 159.06 | 131.37 | 194.74 | −0.44 | −0.47 | −0.42 |
| Yazd | Females | 174.02 | 145.29 | 210.98 | 105.78 | 88.33 | 127.36 | −0.39 | −0.43 | −0.37 |
| Zanjan | Females | 155.04 | 127.61 | 189.14 | 101.03 | 84.81 | 121.27 | −0.35 | −0.39 | −0.32 |
| All-ages prevalence counts estimates | | | | | | | | | | |
| Iran (Islamic Republic of) | Males | 25,980.61 | 21,921.03 | 31,313.86 | 64,913.99 | 54,867.53 | 76,963.05 | 1.5 | 1.36 | 1.63 |
| Alborz | Males | 799.12 | 666.04 | 960.89 | 3,004.47 | 2,512.81 | 3,580.00 | 2.76 | 2.54 | 2.98 |
| Ardebil | Males | 526.13 | 442.47 | 628.02 | 1,066.37 | 896.7 | 1,266.74 | 1.03 | 0.9 | 1.15 |
| Bushehr | Males | 232.98 | 197.84 | 280.1 | 746.01 | 634.99 | 883.77 | 2.2 | 2.03 | 2.37 |
| Chahar Mahaal and Bakhtiari | Males | 308.66 | 261.85 | 369.18 | 658.52 | 554.24 | 787.27 | 1.13 | 1.01 | 1.24 |
| East Azarbayejan | Males | 1,727.24 | 1,439.06 | 2,099.40 | 3,566.44 | 3,006.81 | 4,224.22 | 1.06 | 0.89 | 1.23 |
| Fars | Males | 1,842.56 | 1,541.73 | 2,227.87 | 4,527.43 | 3,782.49 | 5,373.18 | 1.46 | 1.28 | 1.62 |
| Gilan | Males | 1,064.00 | 889.05 | 1,286.54 | 2,381.49 | 2,017.29 | 2,802.77 | 1.24 | 1.08 | 1.4 |
| Golestan | Males | 543.91 | 459.05 | 656.48 | 1,396.89 | 1,169.92 | 1,669.31 | 1.57 | 1.45 | 1.71 |
| Hamadan | Males | 1,037.36 | 870.46 | 1,245.50 | 1,819.30 | 1,531.40 | 2,154.08 | 0.75 | 0.61 | 0.89 |
| Hormozgan | Males | 395.57 | 333.8 | 470.5 | 1,367.40 | 1,133.36 | 1,646.31 | 2.46 | 2.29 | 2.63 |
| Ilam | Males | 233.76 | 196.32 | 281.15 | 623.48 | 518.14 | 745.61 | 1.67 | 1.48 | 1.86 |
| Isfahan | Males | 1,583.45 | 1,347.54 | 1,896.86 | 4,184.50 | 3,512.20 | 4,998.22 | 1.64 | 1.51 | 1.79 |
| Kerman | Males | 771.08 | 646.96 | 936.16 | 2,417.17 | 2,016.64 | 2,890.05 | 2.13 | 1.95 | 2.33 |
| Kermanshah | Males | 1,188.21 | 993.95 | 1,432.18 | 1,974.55 | 1,651.23 | 2,333.74 | 0.66 | 0.54 | 0.78 |
| Khorasan-e-Razavi | Males | 1,833.49 | 1,550.81 | 2,204.80 | 4,647.60 | 3,941.63 | 5,499.90 | 1.53 | 1.37 | 1.7 |
| Khuzestan | Males | 1,194.35 | 1,014.86 | 1,439.53 | 3,419.90 | 2,874.35 | 4,087.69 | 1.86 | 1.72 | 2.02 |
| Kohgiluyeh and Boyer-Ahmad | Males | 219.1 | 182.4 | 270.9 | 646.6 | 532.49 | 777.9 | 1.95 | 1.74 | 2.16 |
| Kurdistan | Males | 578.62 | 491.99 | 692.14 | 1,280.78 | 1,072.53 | 1,526.19 | 1.21 | 1.09 | 1.34 |
| Lorestan | Males | 811.01 | 682.51 | 967.64 | 1,543.82 | 1,288.03 | 1,833.60 | 0.9 | 0.8 | 1.01 |
| Markazi | Males | 617.56 | 518.61 | 748.5 | 1,128.17 | 948.56 | 1,336.84 | 0.83 | 0.7 | 0.96 |
| Mazandaran | Males | 919.91 | 780.92 | 1,100.78 | 2,800.21 | 2,371.82 | 3,320.58 | 2.04 | 1.88 | 2.23 |

(*Continued*)

**Table 5.** (*Continued*)

| Location | Sex | 1990 | | | 2021 | | | Percentage change 1990–2021 | | |
|---|---|---|---|---|---|---|---|---|---|---|
| | | Value | Lower | Upper | Value | Lower | Upper | Value | Lower | Upper |
| North Khorasan | Males | 255.36 | 216.95 | 307.29 | 562.07 | 477.11 | 670.29 | 1.2 | 1.06 | 1.35 |
| Qazvin | Males | 283.28 | 241.67 | 340.29 | 970.2 | 817.79 | 1,146.22 | 2.42 | 2.23 | 2.65 |
| Qom | Males | 309.33 | 260.62 | 375.88 | 992.11 | 825.86 | 1,188.43 | 2.21 | 2 | 2.46 |
| Semnan | Males | 186.59 | 160 | 221.95 | 537.86 | 454.96 | 635.85 | 1.88 | 1.75 | 2.03 |
| Sistan and Baluchistan | Males | 419.79 | 362.38 | 494.65 | 1,494.14 | 1,261.12 | 1,780.24 | 2.56 | 2.42 | 2.73 |
| South Khorasan | Males | 223.13 | 192.09 | 261.35 | 542.42 | 460.96 | 643.26 | 1.43 | 1.31 | 1.56 |
| Tehran | Males | 4,320.45 | 3,699.32 | 5,140.16 | 10,377.91 | 8,885.78 | 12,242.71 | 1.4 | 1.28 | 1.51 |
| West Azarbayejan | Males | 1,044.44 | 873.45 | 1,276.68 | 2,796.59 | 2,335.12 | 3,312.87 | 1.68 | 1.48 | 1.9 |
| Yazd | Males | 245.46 | 210.04 | 292.23 | 751.43 | 634.95 | 893.43 | 2.06 | 1.92 | 2.24 |
| Zanjan | Males | 264.71 | 226.32 | 316.18 | 688.16 | 584.17 | 815.08 | 1.6 | 1.48 | 1.76 |
| Iran (Islamic Republic of) | Females | 39,854.26 | 32,912.27 | 49,170.48 | 57,867.77 | 48,020.96 | 70,468.16 | 0.45 | 0.36 | 0.53 |
| Alborz | Females | 897.71 | 735.44 | 1,117.21 | 2,255.41 | 1,843.73 | 2,758.72 | 1.51 | 1.38 | 1.64 |
| Ardebil | Females | 910.45 | 751.28 | 1,133.08 | 953.21 | 794.49 | 1,149.59 | 0.05 | −0.03 | 0.12 |
| Bushehr | Females | 571.59 | 468.55 | 710.8 | 827.09 | 689.69 | 993.68 | 0.45 | 0.32 | 0.56 |
| Chahar Mahaal and Bakhtiari | Females | 605.73 | 496.13 | 746.28 | 755.37 | 622.38 | 918.81 | 0.25 | 0.15 | 0.33 |
| East Azarbayejan | Females | 2,376.11 | 1,938.71 | 2,969.95 | 2,899.41 | 2,405.31 | 3,533.16 | 0.22 | 0.13 | 0.3 |
| Fars | Females | 3,092.95 | 2,512.98 | 3,879.40 | 4,191.23 | 3,453.43 | 5,132.54 | 0.36 | 0.27 | 0.44 |
| Gilan | Females | 1,579.18 | 1,291.76 | 1,943.30 | 2,058.57 | 1,709.06 | 2,493.13 | 0.3 | 0.23 | 0.38 |
| Golestan | Females | 1,355.35 | 1,099.45 | 1,738.65 | 1,632.84 | 1,356.21 | 1,986.89 | 0.2 | 0.09 | 0.32 |
| Hamadan | Females | 1,304.44 | 1,062.72 | 1,644.82 | 1,413.44 | 1,166.56 | 1,719.03 | 0.08 | 0.01 | 0.16 |
| Hormozgan | Females | 539.4 | 446.17 | 662.17 | 995.01 | 829.73 | 1,212.26 | 0.84 | 0.73 | 0.96 |
| Ilam | Females | 480.97 | 388.86 | 602.86 | 644.05 | 532.2 | 776.04 | 0.34 | 0.23 | 0.45 |
| Isfahan | Females | 1,864.13 | 1,541.57 | 2,294.25 | 3,176.54 | 2,668.91 | 3,838.50 | 0.7 | 0.58 | 0.8 |
| Kerman | Females | 1,172.83 | 966.64 | 1,449.53 | 2,144.63 | 1,771.82 | 2,645.44 | 0.83 | 0.73 | 0.91 |
| Kermanshah | Females | 2,097.00 | 1,707.56 | 2,682.03 | 2,154.17 | 1,763.87 | 2,652.71 | 0.03 | −0.05 | 0.11 |
| Khorasan-e-Razavi | Females | 3,331.09 | 2,728.24 | 4,131.31 | 4,232.83 | 3,510.21 | 5,115.22 | 0.27 | 0.17 | 0.36 |
| Khuzestan | Females | 2,340.69 | 1,917.52 | 2,916.10 | 3,636.86 | 2,971.03 | 4,474.76 | 0.55 | 0.47 | 0.63 |
| Kohgiluyeh and Boyer-Ahmad | Females | 416.87 | 337.4 | 533.86 | 687.17 | 556.58 | 853.49 | 0.65 | 0.54 | 0.76 |
| Kurdistan | Females | 1,148.07 | 937.16 | 1,447.30 | 1,215.41 | 1,010.46 | 1,471.07 | 0.06 | −0.04 | 0.14 |
| Lorestan | Females | 1,554.36 | 1,273.86 | 1,954.09 | 1,804.10 | 1,488.74 | 2,188.69 | 0.16 | 0.08 | 0.25 |
| Markazi | Females | 709.95 | 589.45 | 871.35 | 822.79 | 694.96 | 990.55 | 0.16 | 0.06 | 0.23 |
| Mazandaran | Females | 1,448.63 | 1,184.90 | 1,810.57 | 2,441.21 | 2,001.92 | 2,965.36 | 0.69 | 0.58 | 0.79 |
| North Khorasan | Females | 599.95 | 494.62 | 756.91 | 694.63 | 575.09 | 843.12 | 0.16 | 0.08 | 0.23 |
| Qazvin | Females | 452.71 | 377.39 | 554.24 | 778.39 | 652.81 | 944.25 | 0.72 | 0.62 | 0.81 |
| Qom | Females | 268.75 | 226.52 | 323.73 | 626.73 | 529.19 | 758.85 | 1.33 | 1.2 | 1.46 |
| Semnan | Females | 236.57 | 197.03 | 289.77 | 407.88 | 342 | 493.68 | 0.72 | 0.62 | 0.81 |
| Sistan and Baluchistan | Females | 547.85 | 460.26 | 667.46 | 1,394.56 | 1,160.24 | 1,705.03 | 1.55 | 1.45 | 1.63 |
| South Khorasan | Females | 391.6 | 327.67 | 478.47 | 533.79 | 445.96 | 652.73 | 0.36 | 0.29 | 0.42 |
| Tehran | Females | 4,583.38 | 3,826.56 | 5,549.04 | 8,358.90 | 7,024.47 | 10,099.26 | 0.82 | 0.71 | 0.91 |
| West Azarbayejan | Females | 2,149.33 | 1,741.77 | 2,737.79 | 2,890.52 | 2,378.10 | 3,553.10 | 0.34 | 0.25 | 0.44 |
| Yazd | Females | 390.49 | 324.71 | 477.85 | 630.17 | 523.85 | 762.77 | 0.61 | 0.51 | 0.7 |

(*Continued*)

**Table 5.** (*Continued*)

| Location | Sex | Year | | | | | | | | |
|---|---|---|---|---|---|---|---|---|---|---|
| | | 1990 | | | 2021 | | | Percentage change 1990–2021 | | |
| | | Value | Lower | Upper | Value | Lower | Upper | Value | Lower | Upper |
| Zanjan | Females | 436.14 | 357.65 | 541.26 | 610.84 | 510.35 | 735.01 | 0.4 | 0.29 | 0.49 |
| Age-standardized rate incidence (per 100,000) | | | | | | | | | | |
| Iran (Islamic Republic of) | Males | 39.72 | 32.38 | 48.02 | 42.9 | 33.59 | 53.18 | 0.08 | 0 | 0.15 |
| Alborz | Males | 48.1 | 38.96 | 59.03 | 54.42 | 42.69 | 68.08 | 0.13 | 0.03 | 0.23 |
| Ardebil | Males | 41.99 | 34.41 | 51.28 | 45.82 | 35.69 | 57.3 | 0.09 | −0.02 | 0.19 |
| Bushehr | Males | 31.21 | 25.45 | 38.25 | 36.19 | 28.39 | 44.41 | 0.16 | 0.05 | 0.28 |
| Chahar Mahaal and Bakhtiari | Males | 41.53 | 33.68 | 50.62 | 39.02 | 30.82 | 48.73 | −0.06 | −0.15 | 0.03 |
| East Azarbayejan | Males | 41.09 | 33.28 | 50.27 | 46.58 | 36.29 | 58.7 | 0.13 | 0.02 | 0.27 |
| Fars | Males | 45.45 | 37 | 55.34 | 48.4 | 37.83 | 60.3 | 0.06 | −0.03 | 0.16 |
| Gilan | Males | 40.3 | 32.91 | 48.8 | 45.19 | 35.38 | 56.99 | 0.12 | 0 | 0.23 |
| Golestan | Males | 37.94 | 30.66 | 46.1 | 42.16 | 32.92 | 53.06 | 0.11 | 0.02 | 0.23 |
| Hamadan | Males | 52.45 | 42.43 | 64.45 | 55.56 | 43.28 | 68.82 | 0.06 | −0.04 | 0.16 |
| Hormozgan | Males | 38.5 | 31.11 | 46.72 | 43.79 | 34.36 | 54.93 | 0.14 | 0.03 | 0.24 |
| Ilam | Males | 50.27 | 41.26 | 61.28 | 58.1 | 45 | 72.6 | 0.16 | 0.04 | 0.27 |
| Isfahan | Males | 35.49 | 28.47 | 43.13 | 41.06 | 32.74 | 50.77 | 0.16 | 0.06 | 0.26 |
| Kerman | Males | 36.98 | 29.85 | 44.73 | 41.55 | 32.1 | 52.04 | 0.12 | 0.01 | 0.24 |
| Kermanshah | Males | 61.34 | 50.34 | 74.75 | 53.38 | 41.59 | 66.15 | −0.13 | −0.21 | −0.04 |
| Khorasan-e-Razavi | Males | 33.79 | 27.52 | 40.97 | 40.27 | 31.53 | 50.11 | 0.19 | 0.08 | 0.31 |
| Khuzestan | Males | 36.08 | 28.98 | 44.22 | 41.66 | 32.7 | 51.78 | 0.15 | 0.07 | 0.24 |
| Kohgiluyeh and Boyer-Ahmad | Males | 43.15 | 34.7 | 53.23 | 50.59 | 38.96 | 64.31 | 0.17 | 0.05 | 0.31 |
| Kurdistan | Males | 41.32 | 33.83 | 50.2 | 40.97 | 31.75 | 51.37 | −0.01 | −0.11 | 0.09 |
| Lorestan | Males | 48.83 | 39.9 | 59.32 | 51.73 | 40.34 | 63.26 | 0.06 | −0.03 | 0.15 |
| Markazi | Males | 44 | 35.67 | 52.97 | 40.57 | 31.82 | 51.58 | −0.08 | −0.16 | 0.01 |
| Mazandaran | Males | 31.42 | 25.57 | 38.43 | 41.07 | 32.35 | 51.67 | 0.31 | 0.21 | 0.43 |
| North Khorasan | Males | 34.68 | 28.02 | 42.38 | 38.13 | 30 | 47.22 | 0.1 | −0.01 | 0.2 |
| Qazvin | Males | 26.83 | 21.79 | 32.55 | 39.76 | 31.53 | 49.6 | 0.48 | 0.35 | 0.64 |
| Qom | Males | 36.96 | 30.05 | 44.49 | 40.46 | 31.41 | 50.45 | 0.09 | 0 | 0.2 |
| Semnan | Males | 31.35 | 25.23 | 38.06 | 38.6 | 29.97 | 48.37 | 0.23 | 0.13 | 0.33 |
| Sistan and Baluchistan | Males | 27.74 | 21.92 | 34.25 | 37.47 | 28.93 | 47.15 | 0.35 | 0.25 | 0.46 |
| South Khorasan | Males | 28.03 | 22.76 | 33.88 | 38.28 | 30.21 | 47.44 | 0.37 | 0.24 | 0.49 |
| Tehran | Males | 41.75 | 33.7 | 50.54 | 38.31 | 29.94 | 47.4 | −0.08 | −0.17 | 0 |
| West Azarbayejan | Males | 38.43 | 31.11 | 47.37 | 47.1 | 36.98 | 58.71 | 0.23 | 0.12 | 0.36 |
| Yazd | Males | 30.86 | 25.05 | 37.32 | 36.93 | 28.81 | 46.11 | 0.2 | 0.08 | 0.32 |
| Zanjan | Males | 28.29 | 22.73 | 34.5 | 35.28 | 28.15 | 43.95 | 0.25 | 0.12 | 0.35 |
| Iran (Islamic Republic of) | Females | 75.31 | 60.81 | 92.21 | 49.28 | 38.52 | 62 | −0.35 | −0.39 | −0.31 |
| Alborz | Females | 69.93 | 56.65 | 86.89 | 53.88 | 41.9 | 68.12 | −0.23 | −0.31 | −0.16 |
| Ardebil | Females | 90.98 | 73.02 | 112.32 | 52.35 | 40.71 | 65.42 | −0.42 | −0.48 | −0.37 |
| Bushehr | Females | 93.4 | 75.67 | 114.33 | 55.48 | 43.55 | 69.35 | −0.41 | −0.46 | −0.35 |
| Chahar Mahaal and Bakhtiari | Females | 102.83 | 83.08 | 126.35 | 58.52 | 46.08 | 73.73 | −0.43 | −0.48 | −0.37 |
| East Azarbayejan | Females | 70.01 | 55.34 | 86.87 | 48.49 | 37.42 | 62.21 | −0.31 | −0.38 | −0.24 |
| Fars | Females | 94.77 | 76.48 | 116.55 | 57.71 | 44.92 | 73.28 | −0.39 | −0.44 | −0.33 |

(*Continued*)

**Table 5.** (*Continued*)

| Location | Sex | Year | | | | | | | | |
| | | 1990 | | | 2021 | | | Percentage change 1990–2021 | | |
| | | Value | Lower | Upper | Value | Lower | Upper | Value | Lower | Upper |
|---|---|---|---|---|---|---|---|---|---|---|
| Gilan | Females | 71.35 | 57.66 | 87.03 | 50.4 | 39.7 | 63.99 | −0.29 | −0.36 | −0.23 |
| Golestan | Females | 112.34 | 90.13 | 138.03 | 60.77 | 47.93 | 75.37 | −0.46 | −0.51 | −0.41 |
| Hamadan | Females | 80.68 | 65.52 | 99.61 | 55.05 | 42.47 | 69.87 | −0.32 | −0.39 | −0.25 |
| Hormozgan | Females | 66.64 | 53.85 | 82.32 | 41.61 | 32.16 | 52.86 | −0.38 | −0.43 | −0.3 |
| Ilam | Females | 139.23 | 113.16 | 168.87 | 77.78 | 61.73 | 95.92 | −0.44 | −0.49 | −0.39 |
| Isfahan | Females | 52.12 | 41.67 | 65.11 | 40.98 | 31.81 | 51.61 | −0.21 | −0.29 | −0.15 |
| Kerman | Females | 68.94 | 55.26 | 85.3 | 48.36 | 37.41 | 61.5 | −0.3 | −0.37 | −0.22 |
| Kermanshah | Females | 138.04 | 112.06 | 169.9 | 72.75 | 57.16 | 90.73 | −0.47 | −0.52 | −0.42 |
| Khorasan-e-Razavi | Females | 74.15 | 59.57 | 92.46 | 45.74 | 35.58 | 57.57 | −0.38 | −0.44 | −0.33 |
| Khuzestan | Females | 87.24 | 69.38 | 108.88 | 56.74 | 43.87 | 71.17 | −0.35 | −0.41 | −0.29 |
| Kohgiluyeh and Boyer-Ahmad | Females | 107.69 | 87.03 | 132.6 | 71.09 | 55.83 | 89.33 | −0.34 | −0.4 | −0.28 |
| Kurdistan | Females | 101.26 | 81.7 | 122.94 | 50.66 | 39.87 | 64.11 | −0.5 | −0.55 | −0.45 |
| Lorestan | Females | 119.17 | 96.36 | 147.34 | 75.39 | 59.05 | 93.69 | −0.37 | −0.42 | −0.32 |
| Markazi | Females | 59.36 | 47.63 | 73.22 | 38.29 | 30.12 | 48.37 | −0.35 | −0.41 | −0.29 |
| Mazandaran | Females | 58 | 46.51 | 71.38 | 46.12 | 35.57 | 57.86 | −0.2 | −0.28 | −0.13 |
| North Khorasan | Females | 97.64 | 78.89 | 119.7 | 59.16 | 46.52 | 74.68 | −0.39 | −0.45 | −0.34 |
| Qazvin | Females | 52.13 | 41.71 | 64.24 | 41.25 | 32.29 | 52.19 | −0.21 | −0.29 | −0.13 |
| Qom | Females | 40.48 | 32.78 | 50.57 | 33.92 | 25.83 | 43.45 | −0.16 | −0.26 | −0.08 |
| Semnan | Females | 47.48 | 38.05 | 59.29 | 37.41 | 28.89 | 47.26 | −0.21 | −0.3 | −0.14 |
| Sistan and Baluchistan | Females | 44.55 | 35.73 | 54.64 | 41.62 | 32 | 53.52 | −0.07 | −0.15 | 0.03 |
| South Khorasan | Females | 60.06 | 48.63 | 74.43 | 46.46 | 36.04 | 58.83 | −0.23 | −0.3 | −0.16 |
| Tehran | Females | 56.12 | 45.41 | 69.69 | 40.44 | 31.08 | 51.82 | −0.28 | −0.35 | −0.21 |
| West Azarbayejan | Females | 97.25 | 78.63 | 119.79 | 61.7 | 48.35 | 76.94 | −0.37 | −0.43 | −0.3 |
| Yazd | Females | 60.92 | 48.29 | 74.42 | 41.81 | 32.46 | 53.5 | −0.31 | −0.37 | −0.25 |
| Zanjan | Females | 56.57 | 45.39 | 69.97 | 39.65 | 30.62 | 50.19 | −0.3 | −0.37 | −0.23 |
| All-ages incidence counts estimates | | | | | | | | | | |
| Iran (Islamic Republic of) | Males | 10,863.93 | 8,683.10 | 13,557.19 | 19,891.51 | 15,639.20 | 24,678.36 | 0.83 | 0.61 | 1.09 |
| Alborz | Males | 349.13 | 277.04 | 435.04 | 919.83 | 722.14 | 1,136.48 | 1.63 | 1.3 | 2.04 |
| Ardebil | Males | 227.44 | 178.89 | 286.2 | 335.04 | 260.71 | 417.72 | 0.47 | 0.24 | 0.75 |
| Bushehr | Males | 103.53 | 81.91 | 129.82 | 265.07 | 206.41 | 327.44 | 1.56 | 1.22 | 1.96 |
| Chahar Mahaal and Bakhtiari | Males | 135.96 | 107.26 | 171.83 | 212.83 | 168.2 | 266.23 | 0.57 | 0.35 | 0.82 |
| East Azarbayejan | Males | 696.7 | 552.65 | 887.62 | 1,035.48 | 810.01 | 1,287.90 | 0.49 | 0.26 | 0.77 |
| Fars | Males | 787.85 | 623.31 | 989.99 | 1,385.60 | 1,080.34 | 1,737.76 | 0.76 | 0.52 | 1.04 |
| Gilan | Males | 451.4 | 360.52 | 562.08 | 637.71 | 504 | 794.34 | 0.41 | 0.19 | 0.66 |
| Golestan | Males | 229.91 | 179.83 | 287.42 | 431.8 | 338.18 | 538.01 | 0.88 | 0.64 | 1.13 |
| Hamadan | Males | 439.32 | 344.46 | 555.95 | 523.41 | 408.46 | 649.82 | 0.19 | 0.02 | 0.39 |
| Hormozgan | Males | 160.93 | 126.8 | 200.21 | 480.39 | 370.69 | 606.76 | 1.99 | 1.59 | 2.38 |
| Ilam | Males | 105.95 | 83.72 | 133.72 | 200.72 | 155.85 | 251.21 | 0.89 | 0.6 | 1.23 |
| Isfahan | Males | 671.02 | 528.33 | 836.46 | 1,181.98 | 935.71 | 1,457.97 | 0.76 | 0.51 | 1.05 |
| Kerman | Males | 317.11 | 246.88 | 391.76 | 785.62 | 604.24 | 981.57 | 1.48 | 1.13 | 1.88 |
| Kermanshah | Males | 503.87 | 403.65 | 633.65 | 587.22 | 460.29 | 726.85 | 0.17 | 0 | 0.38 |

(*Continued*)

**Table 5.** (*Continued*)

| Location | Sex | Year | | | | | | | | |
| | | 1990 | | | 2021 | | | Percentage change 1990–2021 | | |
| | | Value | Lower | Upper | Value | Lower | Upper | Value | Lower | Upper |
|---|---|---|---|---|---|---|---|---|---|---|
| Khorasan-e-Razavi | Males | 730.31 | 582.84 | 914.52 | 1,480.67 | 1,164.70 | 1,829.01 | 1.03 | 0.74 | 1.36 |
| Khuzestan | Males | 527.11 | 411.93 | 672.32 | 1,109.54 | 861.75 | 1,382.74 | 1.1 | 0.85 | 1.41 |
| Kohgiluyeh and Boyer-Ahmad | Males | 100.74 | 78.91 | 129.96 | 223.81 | 171.48 | 285.39 | 1.22 | 0.86 | 1.62 |
| Kurdistan | Males | 239.46 | 190.86 | 298.92 | 395.72 | 304.63 | 494.97 | 0.65 | 0.4 | 0.95 |
| Lorestan | Males | 355.86 | 281.4 | 447.12 | 506.42 | 395.45 | 624.25 | 0.42 | 0.25 | 0.65 |
| Markazi | Males | 253.53 | 200.4 | 317.66 | 321.6 | 252.86 | 400.58 | 0.27 | 0.09 | 0.47 |
| Mazandaran | Males | 390.13 | 308.76 | 487.34 | 780.42 | 615.57 | 967.57 | 1 | 0.72 | 1.35 |
| North Khorasan | Males | 102.75 | 81.65 | 129.89 | 175.36 | 136.51 | 218.26 | 0.71 | 0.47 | 0.98 |
| Qazvin | Males | 119.73 | 94.42 | 151.06 | 301.01 | 237.57 | 374.02 | 1.51 | 1.14 | 2 |
| Qom | Males | 132.81 | 105.17 | 163.95 | 316.62 | 245.77 | 394.79 | 1.38 | 1.06 | 1.73 |
| Semnan | Males | 73.01 | 57.85 | 90.05 | 173.62 | 133.57 | 217.61 | 1.38 | 1.1 | 1.7 |
| Sistan and Baluchistan | Males | 169.5 | 130.86 | 215.52 | 595.14 | 452.38 | 764.92 | 2.51 | 2.15 | 2.88 |
| South Khorasan | Males | 81.25 | 64.94 | 100.04 | 181.89 | 142.49 | 226.33 | 1.24 | 0.98 | 1.55 |
| Tehran | Males | 1,752.87 | 1,391.40 | 2,166.57 | 3,005.67 | 2,368.74 | 3,694.62 | 0.71 | 0.51 | 0.95 |
| West Azarbayejan | Males | 441.43 | 347.17 | 562.15 | 889.18 | 698.07 | 1,109.11 | 1.01 | 0.74 | 1.35 |
| Yazd | Males | 99.38 | 78.97 | 125 | 239.72 | 185.31 | 297.94 | 1.41 | 1.12 | 1.75 |
| Zanjan | Males | 113.93 | 89.73 | 142.92 | 212.42 | 168.78 | 265.49 | 0.86 | 0.54 | 1.18 |
| Iran (Islamic Republic of) | Females | 21,219.21 | 16,839.31 | 26,855.28 | 21,017.07 | 16,503.98 | 26,128.95 | −0.01 | −0.11 | 0.12 |
| Alborz | Females | 503.71 | 396.1 | 628.71 | 818.8 | 639.42 | 1,013.54 | 0.63 | 0.43 | 0.87 |
| Ardebil | Females | 498.76 | 389.38 | 629.6 | 339.64 | 268.61 | 422.02 | −0.32 | −0.41 | −0.21 |
| Bushehr | Females | 325.46 | 258.8 | 412.46 | 353.89 | 276.8 | 443.47 | 0.09 | −0.04 | 0.24 |
| Chahar Mahaal and Bakhtiari | Females | 348.91 | 273 | 436.97 | 294.38 | 232 | 367.34 | −0.16 | −0.27 | −0.03 |
| East Azarbayejan | Females | 1,208.35 | 934.52 | 1,536.14 | 954.99 | 740.22 | 1,186.94 | −0.21 | −0.31 | −0.1 |
| Fars | Females | 1,680.46 | 1,330.32 | 2,108.27 | 1,481.23 | 1,167.81 | 1,854.87 | −0.12 | −0.22 | 0.01 |
| Gilan | Females | 838.7 | 668.31 | 1,040.75 | 643.01 | 510.03 | 785.47 | −0.23 | −0.33 | −0.12 |
| Golestan | Females | 732.93 | 577.22 | 929.77 | 594.32 | 468.88 | 729.33 | −0.19 | −0.29 | −0.07 |
| Hamadan | Females | 669.53 | 528.38 | 850.1 | 465.04 | 363.84 | 578.97 | −0.31 | −0.4 | −0.19 |
| Hormozgan | Females | 288.03 | 227.28 | 365.29 | 421.36 | 327.17 | 525.13 | 0.46 | 0.29 | 0.67 |
| Ilam | Females | 293.75 | 234.37 | 372.99 | 249.02 | 197.19 | 307.94 | −0.15 | −0.28 | 0 |
| Isfahan | Females | 974.47 | 765.58 | 1,248.17 | 1,090.00 | 852.83 | 1,341.85 | 0.12 | −0.03 | 0.3 |
| Kerman | Females | 611.74 | 482.85 | 780.36 | 809.32 | 632.07 | 1,023.44 | 0.32 | 0.16 | 0.53 |
| Kermanshah | Females | 1,130.91 | 888.73 | 1,409.42 | 728.65 | 576.29 | 904.08 | −0.36 | −0.44 | −0.26 |
| Khorasan-e-Razavi | Females | 1,698.79 | 1,340.32 | 2,177.33 | 1,576.79 | 1,231.59 | 1,974.26 | −0.07 | −0.19 | 0.04 |
| Khuzestan | Females | 1,316.03 | 1,032.12 | 1,684.08 | 1,408.26 | 1,089.00 | 1,753.13 | 0.07 | −0.06 | 0.21 |
| Kohgiluyeh and Boyer-Ahmad | Females | 256.64 | 201.84 | 328.19 | 282.52 | 219.99 | 354.32 | 0.1 | −0.05 | 0.28 |
| Kurdistan | Females | 614.05 | 482.59 | 767.4 | 446.05 | 354.05 | 556.64 | −0.27 | −0.38 | −0.16 |
| Lorestan | Females | 892.95 | 704.49 | 1,138.94 | 682.78 | 535.33 | 843.42 | −0.24 | −0.34 | −0.12 |
| Markazi | Females | 358.74 | 281.78 | 458.42 | 279.89 | 221.22 | 348.1 | −0.22 | −0.33 | −0.09 |
| Mazandaran | Females | 779.54 | 616.65 | 980.67 | 796.95 | 622.31 | 975.06 | 0.02 | −0.11 | 0.19 |
| North Khorasan | Females | 303.61 | 238.02 | 379.34 | 254.41 | 201.06 | 316.96 | −0.16 | −0.26 | −0.05 |
| Qazvin | Females | 241.75 | 189.38 | 305.89 | 279.86 | 219.89 | 347.65 | 0.16 | 0 | 0.36 |

(*Continued*)

**Table 5.** (*Continued*)

| Location | Sex | 1990 Value | 1990 Lower | 1990 Upper | 2021 Value | 2021 Lower | 2021 Upper | Percentage change 1990–2021 Value | Percentage change 1990–2021 Lower | Percentage change 1990–2021 Upper |
|---|---|---|---|---|---|---|---|---|---|---|
| Qom | Females | 144.86 | 112.99 | 185.03 | 249.94 | 191.72 | 314.54 | 0.73 | 0.49 | 0.98 |
| Semnan | Females | 110.84 | 87.3 | 141.23 | 154.39 | 118.55 | 194.87 | 0.39 | 0.2 | 0.59 |
| Sistan and Baluchistan | Females | 298.76 | 233.08 | 384.36 | 653.6 | 495.96 | 847.86 | 1.19 | 0.97 | 1.45 |
| South Khorasan | Females | 187.09 | 148.51 | 238.84 | 201.48 | 157.76 | 253.72 | 0.08 | −0.03 | 0.2 |
| Tehran | Females | 2,338.70 | 1,866.34 | 2,988.74 | 2,981.26 | 2,316.35 | 3,738.27 | 0.27 | 0.12 | 0.47 |
| West Azarbayejan | Females | 1,138.65 | 893.93 | 1,434.19 | 1,059.09 | 840.84 | 1,303.36 | −0.07 | −0.19 | 0.06 |
| Yazd | Females | 191.87 | 149.79 | 241.73 | 249.24 | 193.39 | 314.46 | 0.3 | 0.16 | 0.45 |
| Zanjan | Females | 240.65 | 189.64 | 307.1 | 216.89 | 168.89 | 269.18 | −0.1 | −0.22 | 0.05 |
| Age-standardized rate DALYs (per 100,000) | | | | | | | | | | |
| Iran (Islamic Republic of) | Males | 394.97 | 339.53 | 445.59 | 325.29 | 277.26 | 357.39 | −0.18 | −0.28 | −0.03 |
| Alborz | Males | 694.07 | 289.42 | 940.95 | 568.33 | 270.5 | 740.65 | −0.18 | −0.44 | 0.29 |
| Ardebil | Males | 437.62 | 324.74 | 559.98 | 427.56 | 283.87 | 514.44 | −0.02 | −0.31 | 0.35 |
| Bushehr | Males | 255.66 | 188.19 | 363.54 | 185.59 | 145.2 | 236.09 | −0.27 | −0.48 | 0.02 |
| Chahar Mahaal and Bakhtiari | Males | 378.01 | 278.62 | 484.44 | 233.23 | 183.4 | 295.33 | −0.38 | −0.56 | −0.08 |
| East Azarbayejan | Males | 529.26 | 367.79 | 700.6 | 400.04 | 291.82 | 502.15 | −0.24 | −0.45 | 0.07 |
| Fars | Males | 427.21 | 321.88 | 568.62 | 452.77 | 340.94 | 578.21 | 0.06 | −0.27 | 0.5 |
| Gilan | Males | 446.55 | 325.39 | 591.94 | 374.08 | 278.67 | 478.36 | −0.16 | −0.41 | 0.18 |
| Golestan | Males | 374.4 | 278.02 | 490.89 | 389.87 | 310.2 | 486.92 | 0.04 | −0.24 | 0.42 |
| Hamadan | Males | 740.03 | 449.09 | 976.54 | 667.57 | 366.81 | 849.16 | −0.1 | −0.37 | 0.23 |
| Hormozgan | Males | 443.71 | 318.94 | 591.52 | 426.21 | 321.3 | 542.79 | −0.04 | −0.33 | 0.42 |
| Ilam | Males | 803.7 | 358.86 | 1,068.51 | 783.79 | 363.66 | 981.94 | −0.02 | −0.3 | 0.39 |
| Isfahan | Males | 226.08 | 166.18 | 348.91 | 227.97 | 177.84 | 302.1 | 0.01 | −0.31 | 0.5 |
| Kerman | Males | 431.89 | 331.87 | 578.97 | 321.77 | 255.42 | 417.4 | −0.25 | −0.47 | 0.09 |
| Kermanshah | Males | 831.54 | 543.57 | 1,084.82 | 643.3 | 361.75 | 802.83 | −0.23 | −0.45 | 0.07 |
| Khorasan-e-Razavi | Males | 380.64 | 283.59 | 508.14 | 280.69 | 222.07 | 351.07 | −0.26 | −0.49 | 0.05 |
| Khuzestan | Males | 327.9 | 248.49 | 444.45 | 332.82 | 265 | 409.69 | 0.01 | −0.26 | 0.43 |
| Kohgiluyeh and Boyer-Ahmad | Males | 550.17 | 389.45 | 714.78 | 538.36 | 386.37 | 669.28 | −0.02 | −0.31 | 0.42 |
| Kurdistan | Males | 396.96 | 283.42 | 525.24 | 313.95 | 244.99 | 393.81 | −0.21 | −0.45 | 0.19 |
| Lorestan | Males | 706.38 | 373.4 | 925.06 | 506.66 | 268.65 | 637.51 | −0.28 | −0.48 | 0.02 |
| Markazi | Males | 420.77 | 313.47 | 544.22 | 286.03 | 224.74 | 361.11 | −0.32 | −0.53 | −0.05 |
| Mazandaran | Males | 213.73 | 158.55 | 329.37 | 225.5 | 178.65 | 310.47 | 0.06 | −0.26 | 0.54 |
| North Khorasan | Males | 310.01 | 232.57 | 425.74 | 279.62 | 224.36 | 350.7 | −0.1 | −0.35 | 0.3 |
| Qazvin | Males | 238.23 | 180.7 | 360.25 | 220.53 | 169.43 | 282.36 | −0.07 | −0.35 | 0.34 |
| Qom | Males | 367.47 | 276.03 | 484.11 | 325.16 | 251.69 | 413.66 | −0.12 | −0.39 | 0.27 |
| Semnan | Males | 289.04 | 219.31 | 384.44 | 192.44 | 148.03 | 256.73 | −0.33 | −0.53 | −0.03 |
| Sistan and Baluchistan | Males | 224.44 | 160.99 | 362.99 | 294.01 | 223.81 | 421.41 | 0.31 | −0.12 | 0.88 |
| South Khorasan | Males | 236.57 | 173.87 | 343.12 | 221.15 | 178.26 | 280.61 | −0.07 | −0.33 | 0.34 |
| Tehran | Males | 252.38 | 184.91 | 333.36 | 163.33 | 127.01 | 219.3 | −0.35 | −0.54 | −0.03 |
| West Azarbayejan | Males | 509.17 | 372.31 | 662.57 | 408.53 | 289.07 | 502.87 | −0.2 | −0.43 | 0.14 |
| Yazd | Males | 221.89 | 164.94 | 358.27 | 173.3 | 129.73 | 239.73 | −0.22 | −0.45 | 0.14 |
| Zanjan | Males | 253.86 | 193.33 | 330.9 | 184.33 | 144.28 | 242.46 | −0.27 | −0.48 | 0.06 |

(*Continued*)

**Table 5.** (*Continued*)

| Location | Sex | Year | | | | | | Percentage change 1990–2021 | | |
|---|---|---|---|---|---|---|---|---|---|---|
| | | 1990 | | | 2021 | | | | | |
| | | Value | Lower | Upper | Value | Lower | Upper | Value | Lower | Upper |
| Iran (Islamic Republic of) | Females | 303.11 | 202.92 | 343.37 | 124.44 | 108.73 | 155.18 | −0.59 | −0.66 | −0.42 |
| Alborz | Females | 273.44 | 153.68 | 390.8 | 139.71 | 98.2 | 182.74 | −0.49 | −0.68 | −0.08 |
| Ardebil | Females | 380.15 | 226.79 | 510.12 | 146.79 | 106.84 | 178.62 | −0.61 | −0.72 | −0.4 |
| Bushehr | Females | 426.38 | 194.12 | 599.11 | 155.23 | 97.02 | 193.26 | −0.64 | −0.75 | −0.42 |
| Chahar Mahaal and Bakhtiari | Females | 425.11 | 178.38 | 591.25 | 132.98 | 71.17 | 182.46 | −0.69 | −0.8 | −0.49 |
| East Azarbayejan | Females | 298.01 | 190.39 | 411.07 | 124.41 | 93.01 | 167.63 | −0.58 | −0.73 | −0.32 |
| Fars | Females | 385.15 | 210.57 | 522.91 | 178.03 | 132.2 | 233.03 | −0.54 | −0.68 | −0.2 |
| Gilan | Females | 276.55 | 171.73 | 378.93 | 110.85 | 84.48 | 144.12 | −0.6 | −0.74 | −0.36 |
| Golestan | Females | 601.83 | 227.7 | 837.06 | 226.11 | 121.39 | 296.84 | −0.62 | −0.75 | −0.42 |
| Hamadan | Females | 364.15 | 203.55 | 488.33 | 154.19 | 112.64 | 196.73 | −0.58 | −0.71 | −0.34 |
| Hormozgan | Females | 262.49 | 178.8 | 363.78 | 85.31 | 64.19 | 140.85 | −0.67 | −0.78 | −0.42 |
| Ilam | Females | 954.67 | 239.66 | 1,313.03 | 338.08 | 122.5 | 432.97 | −0.65 | −0.75 | −0.38 |
| Isfahan | Females | 126.42 | 89.13 | 223.27 | 59.08 | 41.49 | 128.59 | −0.53 | −0.69 | −0.29 |
| Kerman | Females | 321.43 | 218.47 | 460.37 | 137.46 | 101.5 | 206.1 | −0.57 | −0.73 | −0.31 |
| Kermanshah | Females | 733.54 | 273.89 | 1,038.41 | 307.61 | 146.28 | 414.09 | −0.58 | −0.71 | −0.34 |
| Khorasan-e-Razavi | Females | 328.9 | 211.97 | 445.44 | 114.95 | 87.06 | 160.73 | −0.65 | −0.77 | −0.39 |
| Khuzestan | Females | 364.66 | 243.34 | 504.77 | 171.33 | 129.64 | 219.08 | −0.53 | −0.68 | −0.28 |
| Kohgiluyeh and Boyer-Ahmad | Females | 594.83 | 226.95 | 820.87 | 323.95 | 175.16 | 419.69 | −0.46 | −0.62 | −0.17 |
| Kurdistan | Females | 445.34 | 228.6 | 604.68 | 131.16 | 91.25 | 167.77 | −0.71 | −0.8 | −0.52 |
| Lorestan | Females | 740.93 | 186.48 | 1,033.89 | 243.63 | 90.3 | 318.12 | −0.67 | −0.77 | −0.43 |
| Markazi | Females | 169.85 | 115.91 | 290.92 | 59.69 | 42.24 | 141.3 | −0.65 | −0.77 | −0.46 |
| Mazandaran | Females | 182.45 | 126.12 | 267.71 | 90.27 | 67.65 | 148.97 | −0.51 | −0.69 | −0.21 |
| North Khorasan | Females | 437.35 | 262.32 | 584.12 | 196.46 | 142.49 | 252.85 | −0.55 | −0.7 | −0.27 |
| Qazvin | Females | 207.87 | 150.08 | 286.34 | 75.01 | 53.73 | 111.44 | −0.64 | −0.76 | −0.39 |
| Qom | Females | 86.97 | 53.75 | 229.17 | 31.34 | 19.87 | 107.53 | −0.64 | −0.77 | −0.47 |
| Semnan | Females | 139.16 | 99.41 | 210.76 | 46.91 | 32.94 | 95.89 | −0.66 | −0.78 | −0.45 |
| Sistan and Baluchistan | Females | 152.18 | 100.86 | 227.23 | 100.09 | 68.32 | 256.9 | −0.34 | −0.61 | 0.27 |
| South Khorasan | Females | 209.04 | 140.87 | 286.51 | 92.16 | 72.32 | 139.86 | −0.56 | −0.68 | −0.32 |
| Tehran | Females | 83.99 | 52.92 | 187.24 | 49.94 | 34.47 | 118.61 | −0.41 | −0.62 | −0.07 |
| West Azarbayejan | Females | 568.94 | 298.05 | 785.6 | 257.42 | 173.48 | 326.7 | −0.55 | −0.69 | −0.34 |
| Yazd | Females | 177.79 | 129.8 | 246.98 | 65.08 | 46.07 | 111.03 | −0.63 | −0.76 | −0.44 |
| Zanjan | Females | 212.63 | 148.03 | 290.36 | 74.54 | 56.97 | 111.17 | −0.65 | −0.77 | −0.4 |
| All-ages DALYs counts estimates | | | | | | | | | | |
| Iran (Islamic Republic of) | Males | 105,109.67 | 88,800.06 | 118,654.48 | 150,642.85 | 129,386.56 | 165,530.23 | 0.43 | 0.25 | 0.68 |
| Alborz | Males | 4,785.14 | 1,977.87 | 6,546.54 | 9,620.22 | 4,583.93 | 12,427.56 | 1.01 | 0.39 | 2.16 |
| Ardebil | Males | 2,238.31 | 1,566.58 | 2,962.58 | 3,140.48 | 2,135.42 | 3,791.58 | 0.4 | 0.02 | 0.95 |
| Bushehr | Males | 792.73 | 581.01 | 1,105.82 | 1,353.29 | 1,056.52 | 1,733.32 | 0.71 | 0.21 | 1.43 |
| Chahar Mahaal and Bakhtiari | Males | 1,168.91 | 852.52 | 1,513.74 | 1,278.57 | 1,001.65 | 1,637.58 | 0.09 | −0.23 | 0.65 |
| East Azarbayejan | Males | 8,819.41 | 5,859.44 | 11,866.60 | 8,886.08 | 6,519.33 | 11,162.29 | 0.01 | −0.29 | 0.44 |
| Fars | Males | 7,240.32 | 5,363.35 | 9,567.00 | 12,688.20 | 9,659.50 | 16,177.91 | 0.75 | 0.22 | 1.47 |
| Gilan | Males | 4,827.20 | 3,470.78 | 6,426.42 | 5,302.00 | 3,935.33 | 6,774.20 | 0.1 | −0.22 | 0.55 |

(*Continued*)

**Table 5.** (*Continued*)

| Location | Sex | Year | | | | | | | | |
|---|---|---|---|---|---|---|---|---|---|---|
| | | 1990 | | | 2021 | | | Percentage change 1990–2021 | | |
| | | Value | Lower | Upper | Value | Lower | Upper | Value | Lower | Upper |
| Golestan | Males | 2,158.03 | 1,596.81 | 2,788.46 | 3,929.64 | 3,140.74 | 4,904.34 | 0.82 | 0.33 | 1.48 |
| Hamadan | Males | 5,909.33 | 3,462.24 | 7,788.94 | 6,295.59 | 3,462.19 | 7,975.75 | 0.07 | −0.26 | 0.44 |
| Hormozgan | Males | 1,757.61 | 1,257.63 | 2,384.00 | 4,619.52 | 3,488.14 | 5,866.08 | 1.63 | 0.81 | 2.94 |
| Ilam | Males | 1,562.63 | 656.45 | 2,106.58 | 2,700.39 | 1,275.84 | 3,377.95 | 0.73 | 0.23 | 1.41 |
| Isfahan | Males | 4,300.99 | 3,121.74 | 6,405.25 | 6,536.09 | 5,108.41 | 8,657.84 | 0.52 | 0.05 | 1.26 |
| Kerman | Males | 3,543.70 | 2,688.95 | 4,708.34 | 6,014.05 | 4,802.83 | 7,730.71 | 0.7 | 0.2 | 1.44 |
| Kermanshah | Males | 6,617.79 | 4,107.53 | 8,759.14 | 7,130.22 | 4,071.72 | 8,845.19 | 0.08 | −0.24 | 0.5 |
| Khorasan-e-Razavi | Males | 7,922.77 | 5,805.93 | 10,472.49 | 10,315.00 | 8,102.14 | 12,809.88 | 0.3 | −0.09 | 0.84 |
| Khuzestan | Males | 4,573.24 | 3,421.02 | 6,045.35 | 8,760.33 | 7,013.53 | 10,703.06 | 0.92 | 0.39 | 1.68 |
| Kohgiluyeh and Boyer-Ahmad | Males | 1,283.36 | 875.53 | 1,714.53 | 2,415.38 | 1,762.24 | 3,025.18 | 0.88 | 0.31 | 1.76 |
| Kurdistan | Males | 2,213.84 | 1,575.17 | 2,953.90 | 3,032.71 | 2,365.45 | 3,793.67 | 0.37 | −0.05 | 1.06 |
| Lorestan | Males | 4,735.00 | 2,375.03 | 6,287.15 | 5,044.98 | 2,681.32 | 6,399.49 | 0.07 | −0.24 | 0.53 |
| Markazi | Males | 2,384.36 | 1,755.07 | 3,084.16 | 2,283.17 | 1,785.47 | 2,874.05 | −0.04 | −0.32 | 0.36 |
| Mazandaran | Males | 2,627.34 | 1,935.90 | 3,921.82 | 4,261.76 | 3,402.84 | 5,691.96 | 0.62 | 0.15 | 1.37 |
| North Khorasan | Males | 887.39 | 663.14 | 1,252.67 | 1,290.11 | 1,028.98 | 1,638.99 | 0.45 | 0.04 | 1.06 |
| Qazvin | Males | 988.27 | 736.08 | 1,434.88 | 1,677.70 | 1,287.92 | 2,131.81 | 0.7 | 0.2 | 1.45 |
| Qom | Males | 1,269.80 | 952.6 | 1,663.77 | 2,540.10 | 1,979.52 | 3,210.06 | 1 | 0.4 | 1.87 |
| Semnan | Males | 643.45 | 482.9 | 858.41 | 867.68 | 669.82 | 1,149.98 | 0.35 | −0.04 | 0.99 |
| Sistan and Baluchistan | Males | 1,350.61 | 950.06 | 2,021.48 | 4,677.26 | 3,526.83 | 6,687.79 | 2.46 | 1.35 | 4.16 |
| South Khorasan | Males | 665.21 | 479.13 | 944.35 | 1,054.71 | 848.67 | 1,327.53 | 0.59 | 0.14 | 1.27 |
| Tehran | Males | 10,485.67 | 7,640.32 | 13,685.69 | 12,997.11 | 10,090.36 | 17,426.77 | 0.24 | −0.13 | 0.84 |
| West Azarbayejan | Males | 5,702.28 | 4,097.11 | 7,618.92 | 7,695.77 | 5,446.80 | 9,502.46 | 0.35 | −0.05 | 0.95 |
| Yazd | Males | 691.59 | 505.54 | 1,090.36 | 1,107.88 | 837.76 | 1,536.14 | 0.6 | 0.1 | 1.34 |
| Zanjan | Males | 963.33 | 716.54 | 1,284.21 | 1,126.86 | 886.07 | 1,474.55 | 0.17 | −0.18 | 0.72 |
| Iran (Islamic Republic of) | Females | 83,677.70 | 53,304.15 | 95,584.67 | 52,614.69 | 45,850.79 | 66,073.28 | −0.37 | −0.49 | −0.07 |
| Alborz | Females | 1,902.17 | 1,030.34 | 2,777.59 | 2,074.58 | 1,468.43 | 2,719.96 | 0.09 | −0.31 | 1.03 |
| Ardebil | Females | 2,060.16 | 1,172.19 | 2,818.16 | 955.16 | 695.95 | 1,171.13 | −0.54 | −0.68 | −0.27 |
| Bushehr | Females | 1,433.61 | 634.1 | 2,033.64 | 989.4 | 623.31 | 1,229.33 | −0.31 | −0.53 | 0.12 |
| Chahar Mahaal and Bakhtiari | Females | 1,403.85 | 548.8 | 1,991.43 | 680.1 | 367.91 | 918.53 | −0.52 | −0.69 | −0.19 |
| East Azarbayejan | Females | 5,021.42 | 3,137.44 | 6,997.12 | 2,415.13 | 1,825.18 | 3,279.25 | −0.52 | −0.68 | −0.19 |
| Fars | Females | 6,614.35 | 3,453.55 | 9,141.91 | 4,494.10 | 3,334.20 | 5,960.97 | −0.32 | −0.54 | 0.26 |
| Gilan | Females | 3,159.35 | 1,882.83 | 4,445.75 | 1,442.61 | 1,100.67 | 1,878.44 | −0.54 | −0.71 | −0.25 |
| Golestan | Females | 3,865.98 | 1,413.45 | 5,434.44 | 2,256.45 | 1,225.93 | 2,975.43 | −0.42 | −0.62 | −0.06 |
| Hamadan | Females | 2,931.52 | 1,608.62 | 3,991.56 | 1,283.19 | 963.08 | 1,618.84 | −0.56 | −0.7 | −0.3 |
| Hormozgan | Females | 1,111.61 | 736.35 | 1,567.15 | 858.09 | 644.95 | 1,432.46 | −0.23 | −0.5 | 0.43 |
| Ilam | Females | 1,934.65 | 456.1 | 2,695.96 | 1,107.88 | 403.16 | 1,441.32 | −0.43 | −0.6 | 0.03 |
| Isfahan | Females | 2,300.84 | 1,605.67 | 3,919.77 | 1,558.55 | 1,098.17 | 3,480.22 | −0.32 | −0.56 | 0.04 |
| Kerman | Females | 2,756.65 | 1,823.94 | 4,029.96 | 2,290.01 | 1,702.28 | 3,449.73 | −0.17 | −0.48 | 0.37 |
| Kermanshah | Females | 5,899.59 | 2,165.86 | 8,473.20 | 3,026.95 | 1,455.77 | 4,065.35 | −0.49 | −0.65 | −0.14 |
| Khorasan-e-Razavi | Females | 7,289.02 | 4,593.40 | 9,929.28 | 3,928.37 | 2,964.65 | 5,520.13 | −0.46 | −0.65 | −0.01 |
| Khuzestan | Females | 5,332.78 | 3,422.79 | 7,422.09 | 4,189.12 | 3,217.65 | 5,371.54 | −0.21 | −0.48 | 0.22 |

(*Continued*)

**Table 5.** (*Continued*)

| Location | Sex | Year | | | | | | | | |
|---|---|---|---|---|---|---|---|---|---|---|
| | | 1990 | | | 2021 | | | Percentage change 1990–2021 | | |
| | | Value | Lower | Upper | Value | Lower | Upper | Value | Lower | Upper |
| Kohgiluyeh and Boyer-Ahmad | Females | 1,418.10 | 520.69 | 1,949.37 | 1,251.51 | 698.94 | 1,614.59 | −0.12 | −0.41 | 0.46 |
| Kurdistan | Females | 2,671.71 | 1,306.05 | 3,624.85 | 1,156.29 | 829.01 | 1,478.96 | −0.57 | −0.7 | −0.28 |
| Lorestan | Females | 5,397.03 | 1,277.35 | 7,642.97 | 2,246.37 | 850.28 | 2,932.28 | −0.58 | −0.71 | −0.24 |
| Markazi | Females | 991.16 | 668.58 | 1,656.46 | 433.62 | 309.85 | 1,035.31 | −0.56 | −0.71 | −0.3 |
| Mazandaran | Females | 2,368.52 | 1,587.49 | 3,453.79 | 1,567.71 | 1,176.28 | 2,641.40 | −0.34 | −0.59 | 0.06 |
| North Khorasan | Females | 1,340.82 | 771.66 | 1,829.43 | 833.94 | 613.13 | 1,061.11 | −0.38 | −0.58 | 0.02 |
| Qazvin | Females | 932.49 | 651.71 | 1,276.84 | 516.13 | 369.8 | 773.08 | −0.45 | −0.64 | −0.06 |
| Qom | Females | 296.17 | 177.94 | 780.29 | 228.53 | 144.94 | 791.67 | −0.23 | −0.53 | 0.17 |
| Semnan | Females | 314.76 | 220.79 | 462.81 | 194.31 | 136.6 | 401.39 | −0.38 | −0.61 | 0.03 |
| Sistan and Baluchistan | Females | 979.05 | 613.79 | 1,443.46 | 1,556.90 | 1,054.02 | 4,003.73 | 0.59 | −0.09 | 2.55 |
| South Khorasan | Females | 631.96 | 423.94 | 864.7 | 398.22 | 312.79 | 610.75 | −0.37 | −0.54 | 0 |
| Tehran | Females | 3,351.44 | 2,067.81 | 7,478.63 | 3,564.46 | 2,481.47 | 8,516.25 | 0.06 | −0.33 | 0.74 |
| West Azarbayejan | Females | 6,546.79 | 3,219.58 | 9,091.93 | 4,320.23 | 3,012.45 | 5,442.08 | −0.34 | −0.56 | −0.01 |
| Yazd | Females | 539.19 | 392.81 | 755.47 | 385.07 | 271.44 | 662.05 | −0.29 | −0.55 | 0.11 |
| Zanjan | Females | 880.97 | 582.36 | 1,226.49 | 411.7 | 316.38 | 625.77 | −0.53 | −0.7 | −0.17 |

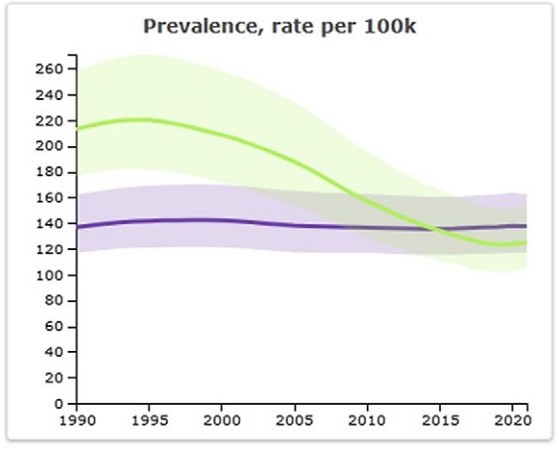

**Figure 5.** Sex-specific prevalence of self-harm in Iran, 1990–2021.

psychological impacts of war-related trauma. Consequently, the rates of self-harm and suicide mortality in 1990 and the subsequent years were significantly elevated. However, over time, these rates transitioned into a sustained downward trajectory, with notable decreases in the prevalence, incidence and disability associated with self-harm, as well as reductions in suicide mortality observed up to 2021.

Studies indicate that war has a profound impact on the mental health of civilians, leading to a noticeable rise in both the prevalence and incidence of mental disorders (Baingana et al., 2005; Lopez-Ibor et al., 2005; Murthy and Lakshminarayana, 2006). A newly published meta-analysis reveals a notable rise in mental health disorders in regions affected by conflict and war (Lim et al., 2022). Wars often lead to a rise in displaced populations, heightened migration, economic instability and the spread of physical illnesses (Lim et al., 2022). Consequently, this can result in a rise in mental health disorders within the community, potentially contributing to an increase in instances of self-harm and suicide-related fatalities.

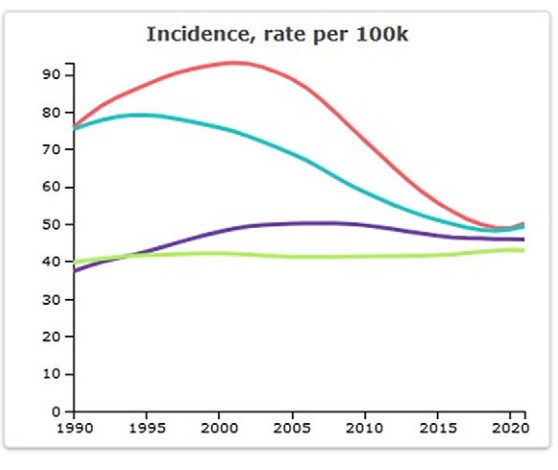

**Figure 6.** Sex-specific incidence of self-harm in Iran, 1990–2021.

**Table 6.** Sex-specific suicide mortality in Iran, stratified by provinces, 1990–2021

| | | Year | | | | | | | | |
| | | 1990 | | | 2021 | | | Percentage change 1990–2021 | | |
| Location | Sex | Value | Lower | Upper | Value | Lower | Upper | Value | Lower | Upper |
|---|---|---|---|---|---|---|---|---|---|---|
| Age-standardized rate suicide mortality (per 100,000) | | | | | | | | | | |
| Iran (Islamic Republic of) | Males | 7.42 | 6.35 | 8.39 | 6.04 | 5.14 | 6.67 | −0.19 | −0.29 | −0.03 |
| Alborz | Males | 13.42 | 5.54 | 18.33 | 10.67 | 4.94 | 13.88 | −0.21 | −0.44 | 0.26 |
| Ardebil | Males | 8.26 | 6.12 | 10.58 | 8.33 | 5.52 | 9.96 | 0.01 | −0.28 | 0.41 |
| Bushehr | Males | 4.94 | 3.66 | 7.21 | 3.69 | 2.9 | 4.71 | −0.25 | −0.47 | 0.04 |
| Chahar Mahaal and Bakhtiari | Males | 7.3 | 5.36 | 9.34 | 4.57 | 3.6 | 5.77 | −0.37 | −0.55 | −0.05 |
| East Azarbayejan | Males | 9.76 | 6.97 | 12.94 | 7.57 | 5.48 | 9.5 | −0.22 | −0.45 | 0.11 |
| Fars | Males | 7.92 | 5.98 | 10.55 | 8.09 | 6.11 | 10.31 | 0.02 | −0.29 | 0.46 |
| Gilan | Males | 8.66 | 6.43 | 11.48 | 7.06 | 5.2 | 8.98 | −0.18 | −0.42 | 0.15 |
| Golestan | Males | 7.09 | 5.29 | 9.49 | 7.21 | 5.75 | 9.1 | 0.02 | −0.26 | 0.4 |
| Hamadan | Males | 13.75 | 8.37 | 18.23 | 12.31 | 6.64 | 15.66 | −0.1 | −0.38 | 0.22 |
| Hormozgan | Males | 8.47 | 6.18 | 11.37 | 7.68 | 5.8 | 9.75 | −0.09 | −0.36 | 0.32 |
| Ilam | Males | 15.56 | 6.84 | 20.51 | 14.67 | 6.6 | 18.34 | −0.06 | −0.32 | 0.34 |
| Isfahan | Males | 4.12 | 3.01 | 6.56 | 4.12 | 3.21 | 5.4 | 0 | −0.32 | 0.5 |
| Kerman | Males | 8.09 | 6.17 | 10.97 | 5.93 | 4.69 | 7.67 | −0.27 | −0.48 | 0.07 |
| Kermanshah | Males | 15.59 | 10.24 | 20.15 | 11.89 | 6.73 | 14.85 | −0.24 | −0.46 | 0.06 |
| Khorasan-e-Razavi | Males | 7.28 | 5.46 | 9.64 | 5.34 | 4.18 | 6.68 | −0.27 | −0.49 | 0.05 |
| Khuzestan | Males | 6.13 | 4.62 | 8.38 | 6.09 | 4.89 | 7.4 | −0.01 | −0.28 | 0.39 |
| Kohgiluyeh and Boyer-Ahmad | Males | 9.57 | 6.86 | 12.39 | 9.35 | 6.73 | 11.6 | −0.02 | −0.31 | 0.39 |
| Kurdistan | Males | 7.49 | 5.36 | 9.98 | 5.77 | 4.52 | 7.28 | −0.23 | −0.47 | 0.16 |
| Lorestan | Males | 13.56 | 7.25 | 17.69 | 10.1 | 5.3 | 12.58 | −0.26 | −0.46 | 0.05 |
| Markazi | Males | 7.74 | 5.72 | 10.18 | 5.22 | 4.1 | 6.63 | −0.33 | −0.52 | −0.03 |
| Mazandaran | Males | 3.94 | 2.9 | 6.3 | 4.15 | 3.3 | 5.64 | 0.05 | −0.26 | 0.53 |

*(Continued)*

**Table 6.**  (*Continued*)

| Location | Sex | 1990 | | | 2021 | | | Percentage change 1990–2021 | | |
|---|---|---|---|---|---|---|---|---|---|---|
| | | Value | Lower | Upper | Value | Lower | Upper | Value | Lower | Upper |
| North Khorasan | Males | 5.79 | 4.34 | 8.22 | 5.16 | 4.14 | 6.48 | −0.11 | −0.36 | 0.3 |
| Qazvin | Males | 4.63 | 3.48 | 7.16 | 4.3 | 3.35 | 5.47 | −0.07 | −0.35 | 0.33 |
| Qom | Males | 6.96 | 5.26 | 9.26 | 5.96 | 4.6 | 7.54 | −0.14 | −0.42 | 0.21 |
| Semnan | Males | 5.59 | 4.21 | 7.53 | 3.71 | 2.87 | 4.89 | −0.34 | −0.53 | −0.03 |
| Sistan and Baluchistan | Males | 4.22 | 3.02 | 7.45 | 5.28 | 4 | 7.7 | 0.25 | −0.15 | 0.82 |
| South Khorasan | Males | 4.45 | 3.28 | 6.51 | 4.13 | 3.32 | 5.23 | −0.07 | −0.35 | 0.35 |
| Tehran | Males | 4.76 | 3.48 | 6.41 | 3.11 | 2.4 | 4.14 | −0.35 | −0.54 | −0.01 |
| West Azarbayejan | Males | 9.49 | 6.91 | 12.27 | 7.7 | 5.45 | 9.44 | −0.19 | −0.42 | 0.15 |
| Yazd | Males | 4.28 | 3.17 | 7.09 | 3.22 | 2.39 | 4.49 | −0.25 | −0.47 | 0.09 |
| Zanjan | Males | 4.89 | 3.72 | 6.53 | 3.6 | 2.85 | 4.76 | −0.26 | −0.47 | 0.06 |
| Iran (Islamic Republic of) | Females | 5.04 | 3.47 | 5.71 | 2.14 | 1.85 | 2.74 | −0.58 | −0.65 | −0.4 |
| Alborz | Females | 4.71 | 2.58 | 6.69 | 2.49 | 1.74 | 3.27 | −0.47 | −0.66 | 0.01 |
| Ardebil | Females | 6.29 | 3.65 | 8.47 | 2.59 | 1.88 | 3.2 | −0.59 | −0.71 | −0.35 |
| Bushehr | Females | 7.35 | 3.15 | 10.38 | 2.86 | 1.73 | 3.55 | −0.61 | −0.73 | −0.37 |
| Chahar Mahaal and Bakhtiari | Females | 7.21 | 2.78 | 9.91 | 2.38 | 1.18 | 3.23 | −0.67 | −0.78 | −0.45 |
| East Azarbayejan | Females | 4.89 | 3.13 | 6.83 | 2.1 | 1.55 | 2.87 | −0.57 | −0.72 | −0.27 |
| Fars | Females | 6.45 | 3.53 | 8.82 | 3.02 | 2.22 | 4.01 | −0.53 | −0.69 | −0.19 |
| Gilan | Females | 4.78 | 2.98 | 6.55 | 1.99 | 1.48 | 2.61 | −0.58 | −0.73 | −0.33 |
| Golestan | Females | 10.02 | 3.54 | 13.97 | 4.1 | 2.09 | 5.35 | −0.59 | −0.72 | −0.36 |
| Hamadan | Females | 6.03 | 3.22 | 8.16 | 2.61 | 1.93 | 3.33 | −0.57 | −0.7 | −0.31 |
| Hormozgan | Females | 4.3 | 2.84 | 6.02 | 1.48 | 1.08 | 2.62 | −0.66 | −0.77 | −0.38 |
| Ilam | Females | 16.74 | 3.82 | 23.06 | 6.54 | 2.21 | 8.38 | −0.61 | −0.72 | −0.29 |
| Isfahan | Females | 2.05 | 1.4 | 3.89 | 0.98 | 0.66 | 2.34 | −0.52 | −0.69 | −0.27 |
| Kerman | Females | 5.36 | 3.61 | 7.65 | 2.33 | 1.71 | 3.69 | −0.57 | −0.73 | −0.29 |
| Kermanshah | Females | 12.39 | 4.33 | 17.53 | 5.33 | 2.43 | 7.17 | −0.57 | −0.71 | −0.3 |
| Khorasan-e-Razavi | Females | 5.44 | 3.49 | 7.41 | 1.98 | 1.5 | 2.77 | −0.64 | −0.76 | −0.36 |
| Khuzestan | Females | 6.12 | 4.09 | 8.5 | 2.94 | 2.27 | 3.78 | −0.52 | −0.67 | −0.25 |
| Kohgiluyeh and Boyer-Ahmad | Females | 9.77 | 3.55 | 13.42 | 5.37 | 2.89 | 6.96 | −0.45 | −0.61 | −0.13 |
| Kurdistan | Females | 7.28 | 3.58 | 10.06 | 2.34 | 1.61 | 3.04 | −0.68 | −0.78 | −0.47 |
| Lorestan | Females | 12.72 | 2.92 | 17.86 | 4.46 | 1.48 | 5.85 | −0.65 | −0.76 | −0.39 |
| Markazi | Females | 2.74 | 1.84 | 5.02 | 0.98 | 0.68 | 2.56 | −0.64 | −0.77 | −0.43 |
| Mazandaran | Females | 3.05 | 2.07 | 4.58 | 1.55 | 1.14 | 2.7 | −0.49 | −0.68 | −0.19 |
| North Khorasan | Females | 7.22 | 4.35 | 9.72 | 3.39 | 2.48 | 4.36 | −0.53 | −0.68 | −0.2 |
| Qazvin | Females | 3.48 | 2.51 | 4.82 | 1.3 | 0.93 | 2.01 | −0.63 | −0.76 | −0.35 |
| Qom | Females | 1.41 | 0.83 | 4.09 | 0.52 | 0.31 | 2.05 | −0.63 | −0.78 | −0.44 |
| Semnan | Females | 2.28 | 1.59 | 3.71 | 0.78 | 0.53 | 1.72 | −0.66 | −0.79 | −0.42 |
| Sistan and Baluchistan | Females | 2.47 | 1.58 | 4.04 | 1.61 | 1.06 | 4.46 | −0.35 | −0.62 | 0.2 |
| South Khorasan | Females | 3.48 | 2.36 | 4.77 | 1.58 | 1.22 | 2.45 | −0.55 | −0.69 | −0.3 |
| Tehran | Females | 1.37 | 0.81 | 3.29 | 0.82 | 0.55 | 2.1 | −0.4 | −0.63 | −0.07 |
| West Azarbayejan | Females | 9.46 | 4.98 | 13.12 | 4.38 | 3 | 5.59 | −0.54 | −0.68 | −0.32 |
| Yazd | Females | 2.98 | 2.12 | 4.2 | 1.1 | 0.77 | 1.99 | −0.63 | −0.76 | −0.43 |
| Zanjan | Females | 3.55 | 2.46 | 4.77 | 1.3 | 0.98 | 2.03 | −0.63 | −0.76 | −0.38 |

**Table 6.** (*Continued*)

| Location | Sex | Year | | | | | | | | |
|---|---|---|---|---|---|---|---|---|---|---|
| | | 1990 | | | 2021 | | | Percentage change 1990–2021 | | |
| | | Value | Lower | Upper | Value | Lower | Upper | Value | Lower | Upper |
| All-ages counts estimates | | | | | | | | | | |
| Iran (Islamic Republic of) | Males | 1,779.40 | 1,522.33 | 2,017.04 | 2,802.33 | 2,398.13 | 3,090.06 | 0.57 | 0.38 | 0.87 |
| Alborz | Males | 82.52 | 33.68 | 112.88 | 181.11 | 84.22 | 235.1 | 1.19 | 0.52 | 2.43 |
| Ardebil | Males | 37.7 | 26.98 | 49.02 | 60.47 | 40.73 | 73.02 | 0.6 | 0.16 | 1.24 |
| Bushehr | Males | 13.5 | 9.83 | 19.07 | 25.43 | 19.86 | 32.9 | 0.88 | 0.36 | 1.67 |
| Chahar Mahaal and Bakhtiari | Males | 20.06 | 14.73 | 25.74 | 24.59 | 19.31 | 31.53 | 0.23 | −0.13 | 0.85 |
| East Azarbayejan | Males | 146.65 | 99.22 | 196.33 | 169.72 | 123.17 | 213.62 | 0.16 | −0.18 | 0.68 |
| Fars | Males | 120.67 | 90.2 | 159.89 | 228.59 | 173.45 | 290.49 | 0.89 | 0.33 | 1.68 |
| Gilan | Males | 84.84 | 61.25 | 113.09 | 104.35 | 76.56 | 132.17 | 0.23 | −0.13 | 0.73 |
| Golestan | Males | 36.67 | 27.04 | 48.18 | 72.12 | 57.49 | 90.58 | 0.97 | 0.43 | 1.7 |
| Hamadan | Males | 99.5 | 59 | 131.23 | 119.22 | 64.24 | 151.64 | 0.2 | −0.17 | 0.64 |
| Hormozgan | Males | 30.28 | 21.58 | 40.85 | 80.55 | 60.46 | 101.58 | 1.66 | 0.85 | 3.02 |
| Ilam | Males | 26.7 | 11.28 | 35.64 | 49.62 | 22.89 | 62.18 | 0.86 | 0.33 | 1.62 |
| Isfahan | Males | 70.69 | 51.48 | 108.4 | 121.2 | 94.64 | 159.93 | 0.71 | 0.17 | 1.53 |
| Kerman | Males | 60 | 45.49 | 80.58 | 108.66 | 85.91 | 140.07 | 0.81 | 0.27 | 1.64 |
| Kermanshah | Males | 111.58 | 70.41 | 146.73 | 133.1 | 75.68 | 165.75 | 0.19 | −0.15 | 0.68 |
| Khorasan-e-Razavi | Males | 136.15 | 100.55 | 182.87 | 193.01 | 150.75 | 239.63 | 0.42 | −0.01 | 1 |
| Khuzestan | Males | 76.57 | 57.33 | 103.55 | 158.01 | 127.39 | 191.2 | 1.06 | 0.5 | 1.89 |
| Kohgiluyeh and Boyer-Ahmad | Males | 20.23 | 14.03 | 26.55 | 41.58 | 30.32 | 52 | 1.06 | 0.44 | 2 |
| Kurdistan | Males | 37.57 | 26.45 | 50.24 | 55.57 | 43.37 | 70.26 | 0.48 | 0.01 | 1.25 |
| Lorestan | Males | 81.28 | 41.66 | 107.26 | 97.83 | 51.14 | 123.76 | 0.2 | −0.15 | 0.71 |
| Markazi | Males | 39.89 | 29.49 | 51.65 | 42.59 | 33.34 | 54.4 | 0.07 | −0.24 | 0.52 |
| Mazandaran | Males | 43.78 | 31.84 | 67.74 | 80.84 | 64.1 | 108.27 | 0.85 | 0.31 | 1.71 |
| North Khorasan | Males | 14.97 | 11.18 | 20.68 | 23.6 | 18.8 | 29.88 | 0.58 | 0.11 | 1.26 |
| Qazvin | Males | 17.05 | 12.75 | 25.37 | 32.35 | 24.8 | 41.1 | 0.9 | 0.34 | 1.74 |
| Qom | Males | 21.34 | 16.02 | 28.15 | 46.05 | 35.76 | 58.16 | 1.16 | 0.49 | 2.16 |
| Semnan | Males | 11.33 | 8.53 | 15.14 | 16.38 | 12.63 | 21.61 | 0.45 | 0.03 | 1.14 |
| Sistan and Baluchistan | Males | 22.87 | 16.19 | 36.01 | 79.56 | 59.88 | 114.86 | 2.48 | 1.31 | 4.18 |
| South Khorasan | Males | 11.56 | 8.39 | 16.9 | 19.29 | 15.55 | 24.44 | 0.67 | 0.18 | 1.39 |
| Tehran | Males | 180.39 | 130.55 | 240.71 | 252.6 | 194.38 | 338.41 | 0.4 | −0.04 | 1.09 |
| West Azarbayejan | Males | 94.39 | 68.78 | 124.41 | 142.16 | 100.73 | 176.06 | 0.51 | 0.06 | 1.17 |
| Yazd | Males | 11.97 | 8.82 | 19.34 | 20.3 | 15.31 | 28.42 | 0.7 | 0.17 | 1.48 |
| Zanjan | Males | 16.7 | 12.55 | 22.13 | 21.85 | 17.21 | 28.7 | 0.31 | −0.07 | 0.89 |
| Iran (Islamic Republic of) | Females | 1,290.26 | 820.94 | 1,468.85 | 906.23 | 779.55 | 1,182.62 | −0.3 | −0.43 | 0.04 |
| Alborz | Females | 29.73 | 15.9 | 43.27 | 35.98 | 24.9 | 48.23 | 0.21 | −0.26 | 1.39 |
| Ardebil | Females | 31.41 | 17.52 | 42.92 | 17.14 | 12.48 | 21.23 | −0.45 | −0.62 | −0.12 |
| Bushehr | Females | 22.43 | 9.57 | 32.18 | 17.49 | 10.71 | 21.9 | −0.22 | −0.47 | 0.3 |
| Chahar Mahaal and Bakhtiari | Females | 21.77 | 8.15 | 31.02 | 12.2 | 6.09 | 16.57 | −0.44 | −0.64 | −0.03 |
| East Azarbayejan | Females | 76.23 | 46.51 | 106.41 | 40.97 | 30.08 | 57.76 | −0.46 | −0.66 | −0.06 |
| Fars | Females | 102.51 | 52.84 | 141.62 | 76.9 | 56.31 | 101.42 | −0.25 | −0.5 | 0.42 |
| Gilan | Females | 50.56 | 30.4 | 70.31 | 27.02 | 20.1 | 35.61 | −0.47 | −0.67 | −0.1 |
| Golestan | Females | 59.63 | 20.74 | 83.94 | 40.82 | 21.09 | 53.83 | −0.32 | −0.55 | 0.11 |

**Table 6.** (*Continued*)

| Location | Sex | Year | | | | | | | | |
| | | 1990 | | | 2021 | | | Percentage change 1990–2021 | | |
| | | Value | Lower | Upper | Value | Lower | Upper | Value | Lower | Upper |
|---|---|---|---|---|---|---|---|---|---|---|
| Hamadan | Females | 45.16 | 24.1 | 61.99 | 22.37 | 16.4 | 28.08 | −0.5 | −0.67 | −0.19 |
| Hormozgan | Females | 16.96 | 11.04 | 24 | 14.27 | 10.33 | 25.47 | −0.16 | −0.46 | 0.58 |
| Ilam | Females | 30.44 | 6.76 | 42.42 | 21.16 | 7.34 | 27.69 | −0.3 | −0.51 | 0.27 |
| Isfahan | Females | 34.73 | 23.73 | 63.22 | 25.92 | 17.38 | 64.09 | −0.25 | −0.54 | 0.17 |
| Kerman | Females | 42.64 | 28.13 | 62.27 | 38.15 | 27.94 | 61.45 | −0.11 | −0.45 | 0.48 |
| Kermanshah | Females | 91.75 | 32.05 | 132.64 | 53.57 | 24.88 | 71.51 | −0.42 | −0.61 | 0 |
| Khorasan-e-Razavi | Females | 111.84 | 70.05 | 153.61 | 66.83 | 50.25 | 94.77 | −0.4 | −0.61 | 0.1 |
| Khuzestan | Females | 82.8 | 52.89 | 117.11 | 70.88 | 54.94 | 92.86 | −0.14 | −0.43 | 0.35 |
| Kohgiluyeh and Boyer-Ahmad | Females | 21.36 | 7.56 | 29.49 | 20.67 | 11.29 | 27 | −0.03 | −0.35 | 0.6 |
| Kurdistan | Females | 40.52 | 19.06 | 55.57 | 20.66 | 14.58 | 26.79 | −0.49 | −0.66 | −0.11 |
| Lorestan | Females | 84.45 | 18.4 | 119.71 | 41.3 | 13.97 | 54.39 | −0.51 | −0.67 | −0.12 |
| Markazi | Females | 14.98 | 10.03 | 26.49 | 7.33 | 5.06 | 19.61 | −0.51 | −0.69 | −0.19 |
| Mazandaran | Females | 36.63 | 24.02 | 54.47 | 27.81 | 20.4 | 49.95 | −0.24 | −0.54 | 0.22 |
| North Khorasan | Females | 20.54 | 11.74 | 27.95 | 14.32 | 10.5 | 18.6 | −0.3 | −0.54 | 0.17 |
| Qazvin | Females | 14.41 | 10.14 | 19.95 | 8.94 | 6.4 | 14.01 | −0.38 | −0.61 | 0.1 |
| Qom | Females | 4.41 | 2.54 | 12.46 | 3.61 | 2.12 | 14.4 | −0.18 | −0.54 | 0.32 |
| Semnan | Females | 4.88 | 3.34 | 7.71 | 3.22 | 2.15 | 7.12 | −0.34 | −0.61 | 0.14 |
| Sistan and Baluchistan | Females | 14.75 | 9.32 | 22.19 | 23.98 | 15.76 | 65.51 | 0.62 | −0.09 | 2.53 |
| South Khorasan | Females | 9.77 | 6.52 | 13.48 | 6.79 | 5.27 | 10.69 | −0.3 | −0.52 | 0.1 |
| Tehran | Females | 50.82 | 29.55 | 122.56 | 59.5 | 38.87 | 155.34 | 0.17 | −0.3 | 0.96 |
| West Azarbayejan | Females | 100.16 | 49.29 | 139.81 | 72.76 | 50.63 | 92.15 | −0.27 | −0.51 | 0.12 |
| Yazd | Females | 8.42 | 5.98 | 11.93 | 6.39 | 4.42 | 11.53 | −0.24 | −0.53 | 0.21 |
| Zanjan | Females | 13.57 | 9.2 | 18.95 | 7.28 | 5.45 | 11.52 | −0.46 | −0.65 | −0.05 |

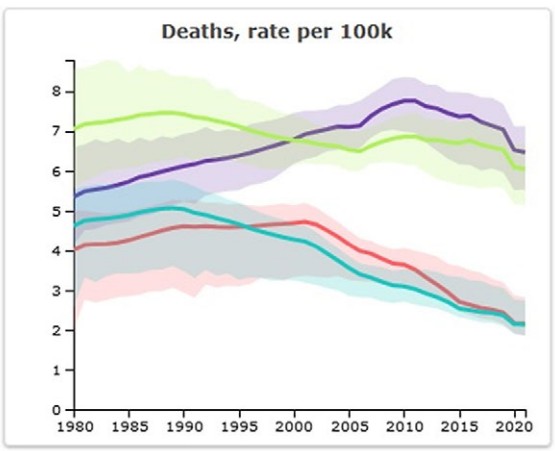

**Figure 7.** Sex-specific suicide mortality in Iran, 1990–2021.

**Table 7.** Age-specific suicide mortality in Iran, stratified by provinces, 1990–2021

| | | Year | | | | | | | | |
| | | 1990 | | | 2021 | | | Percentage change 1990–2021 | | |
| Age | Sex | Value | Lower | Upper | Value | Lower | Upper | Value | Lower | Upper |
|---|---|---|---|---|---|---|---|---|---|---|
| Age— rate suicide mortality (per 100,000) | | | | | | | | | | |
| 10—14 years | Both sexes | 1.99 | 1.26 | 2.36 | 0.66 | 0.55 | 0.81 | −0.67 | −0.74 | −0.51 |
| 15—39 years | Both sexes | 10.46 | 8.07 | 11.6 | 6.82 | 6.09 | 7.48 | −0.35 | −0.43 | −0.19 |
| 40—44 years | Both sexes | 7.15 | 5.87 | 8.26 | 4.83 | 4.3 | 5.55 | −0.32 | −0.42 | −0.1 |
| 45—49 years | Both sexes | 6.76 | 5.65 | 7.76 | 4.34 | 3.85 | 5.09 | −0.36 | −0.45 | −0.13 |
| 50—69 years | Both sexes | 6.26 | 5.39 | 6.99 | 3.77 | 3.37 | 4.38 | −0.4 | −0.46 | −0.23 |
| 70+ years | Both sexes | 6.35 | 5.63 | 7.09 | 4.89 | 4.26 | 5.67 | −0.23 | −0.31 | −0.06 |
| 10—14 years | Males | 2.02 | 1.26 | 2.49 | 0.71 | 0.55 | 0.94 | −0.65 | −0.74 | −0.46 |
| 15—39 years | Males | 11.28 | 9.44 | 12.95 | 10.01 | 8.57 | 11.06 | −0.11 | −0.24 | 0.05 |
| 40—44 years | Males | 9.41 | 7.51 | 11.14 | 7.53 | 6.37 | 8.65 | −0.2 | −0.34 | 0.06 |
| 45—49 years | Males | 9.36 | 7.73 | 11.13 | 6.71 | 5.73 | 7.88 | −0.28 | −0.41 | −0.01 |
| 50—69 years | Males | 8.87 | 7.57 | 10.02 | 5.96 | 5.08 | 6.83 | −0.33 | −0.42 | −0.14 |
| 70+ years | Males | 8 | 7.02 | 9.41 | 6.81 | 5.8 | 7.86 | −0.15 | −0.27 | 0.04 |
| 10—14 years | Females | 1.96 | 1.03 | 2.44 | 0.61 | 0.47 | 0.76 | −0.69 | −0.76 | −0.52 |
| 15—39 years | Females | 9.61 | 5.92 | 11.14 | 3.51 | 3 | 4.42 | −0.63 | −0.72 | −0.44 |
| 40—44 years | Females | 4.84 | 3.55 | 5.61 | 2.05 | 1.71 | 2.79 | −0.58 | −0.66 | −0.36 |
| 45—49 years | Females | 4.1 | 3.01 | 5.02 | 1.85 | 1.56 | 2.78 | −0.55 | −0.64 | −0.33 |
| 50—69 years | Females | 3.25 | 2.49 | 3.77 | 1.58 | 1.34 | 2.29 | −0.52 | −0.6 | −0.33 |
| 70+ years | Females | 4.64 | 3.61 | 5.63 | 3.03 | 2.54 | 4.09 | −0.35 | −0.45 | −0.18 |
| Age count estimates | | | | | | | | | | |
| 10—14 years | Both sexes | 144.58 | 91.89 | 171.23 | 43.7 | 36.68 | 53.76 | −0.7 | −0.76 | −0.55 |
| 15—39 years | Both sexes | 2,270.81 | 1,752.85 | 2,519.34 | 2,367.98 | 2,112.17 | 2,594.32 | 0.04 | −0.09 | 0.3 |
| 40—44 years | Both sexes | 155.59 | 127.84 | 179.74 | 344.75 | 307.21 | 396.52 | 1.22 | 0.91 | 1.95 |
| 45—49 years | Both sexes | 110.47 | 92.3 | 126.89 | 240.73 | 213.44 | 282.62 | 1.18 | 0.87 | 1.95 |
| 50—69 years | Both sexes | 326.58 | 280.94 | 364.8 | 532.74 | 475.98 | 618.61 | 0.63 | 0.46 | 1.09 |
| 70+ years | Both sexes | 61.63 | 54.63 | 68.8 | 178.66 | 155.69 | 207.09 | 1.9 | 1.61 | 2.52 |
| 10—14 years | Males | 74.96 | 46.65 | 92.31 | 24.02 | 18.61 | 31.84 | −0.68 | −0.76 | −0.51 |
| 15—39 years | Males | 1,236.31 | 1,034.02 | 1,418.77 | 1,770.98 | 1,515.40 | 1,957.52 | 0.43 | 0.23 | 0.69 |
| 40—44 years | Males | 103.3 | 82.38 | 122.23 | 272.67 | 230.54 | 313.32 | 1.64 | 1.17 | 2.49 |
| 45—49 years | Males | 77.35 | 63.89 | 92.02 | 190.51 | 162.71 | 223.73 | 1.46 | 1.04 | 2.41 |
| 50—69 years | Males | 247.96 | 211.57 | 280.17 | 421.68 | 359.75 | 483.25 | 0.7 | 0.46 | 1.18 |
| 70+ years | Males | 39.51 | 34.7 | 46.47 | 122.46 | 104.41 | 141.45 | 2.1 | 1.65 | 2.8 |
| 10—14 years | Females | 69.62 | 36.64 | 86.84 | 19.68 | 15.19 | 24.7 | −0.72 | −0.79 | −0.57 |
| 15—39 years | Females | 1,034.50 | 636.5 | 1,199.05 | 596.99 | 509.97 | 751.49 | −0.42 | −0.55 | −0.12 |
| 40—44 years | Females | 52.29 | 38.33 | 60.57 | 72.08 | 60.19 | 98.39 | 0.38 | 0.1 | 1.07 |
| 45—49 years | Females | 33.12 | 24.32 | 40.59 | 50.22 | 42.21 | 75.52 | 0.52 | 0.19 | 1.25 |
| 50—69 years | Females | 78.62 | 60.26 | 91.32 | 111.05 | 94.64 | 161.64 | 0.41 | 0.16 | 0.94 |
| 70+ years | Females | 22.11 | 17.21 | 26.82 | 56.2 | 47.14 | 75.68 | 1.54 | 1.13 | 2.19 |

During the past decades, Iran has been affected by economic turmoil and international politics (Danaei et al., 2019). During the 1980s, Iran witnessed a marked increase in the birth rate, which led to a significant increase in population growth. This increase in population growth affected all aspects of the health system, so there was an increased demand for health infrastructure, health care and

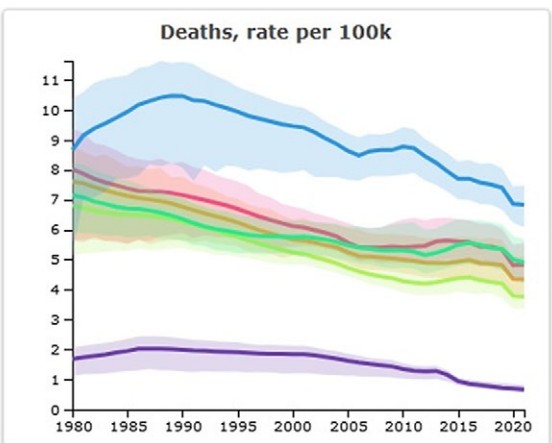

**Legend**

- Iran (Islamic Republic of), Both sexes, 10-14 years, Self-harm
- Iran (Islamic Republic of), Both sexes, 40-44 years, Self-harm
- Iran (Islamic Republic of), Both sexes, 45-49 years, Self-harm
- Iran (Islamic Republic of), Both sexes, 50-69 years, Self-harm
- Iran (Islamic Republic of), Both sexes, 70+ years, Self-harm
- Iran (Islamic Republic of), Both sexes, 15-39 years, Self-harm

**Figure 8.** Age-specific suicide mortality in Iran, 1990–2021.

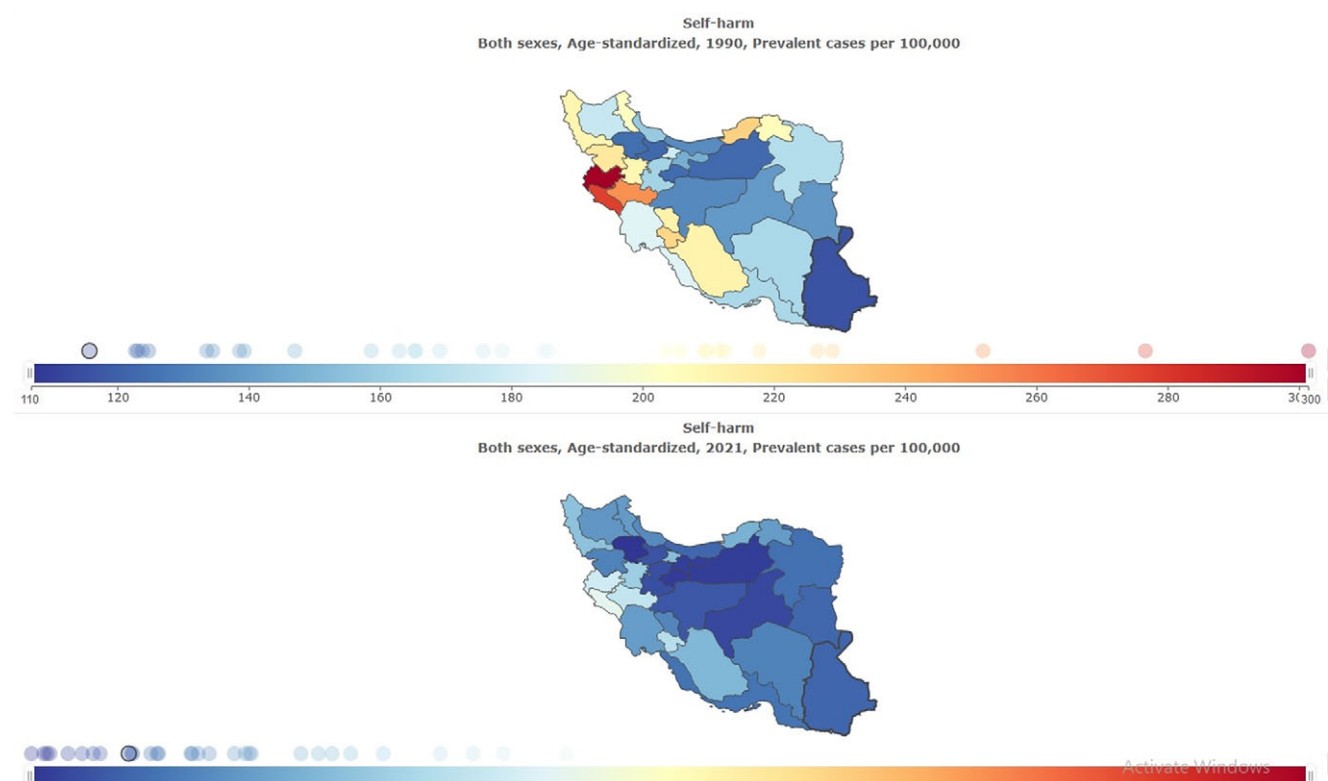

**Figure 9.** Age-standardized rate (per 100,000) prevalence of self-harm in Iran, stratified by provinces, 1990–2021.

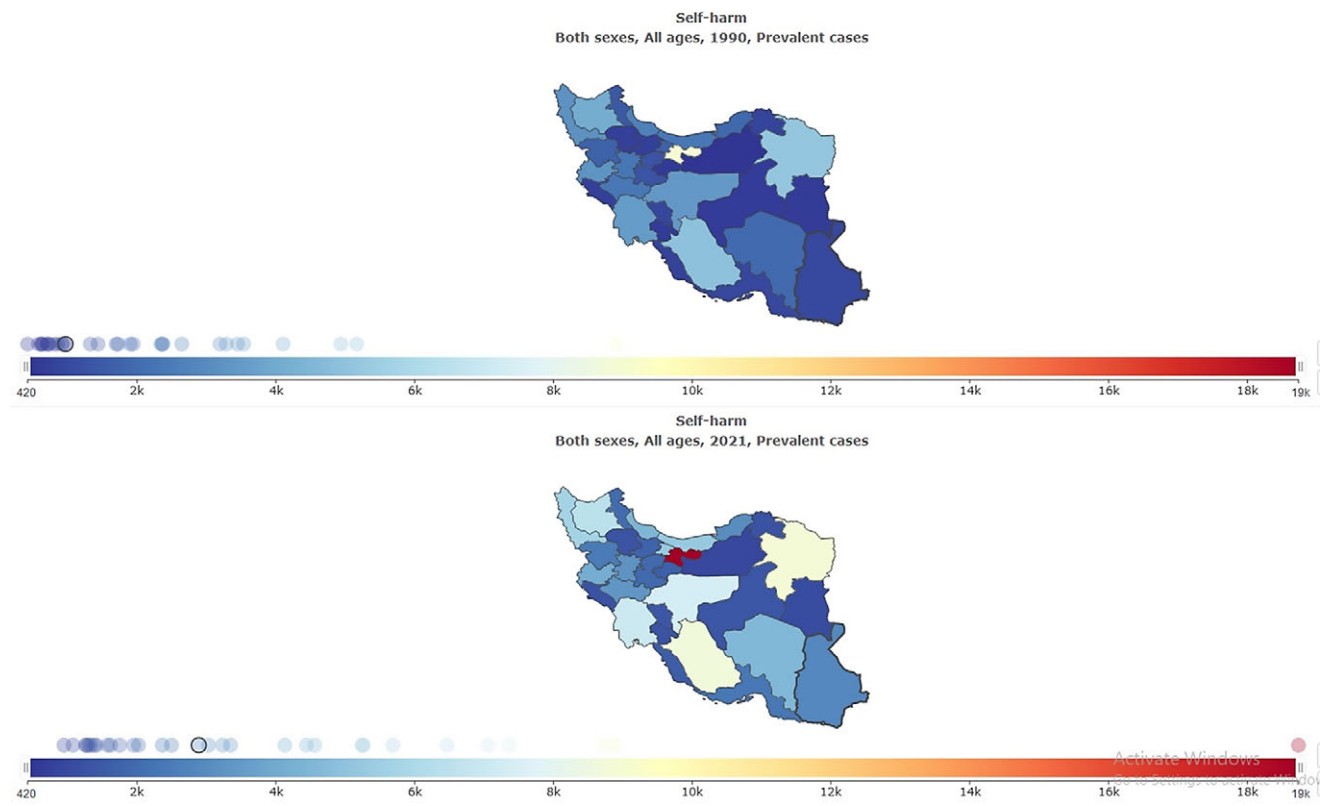

**Figure 10.** All-ages prevalence of self-harm in Iran, stratified by provinces, 1990–2021.

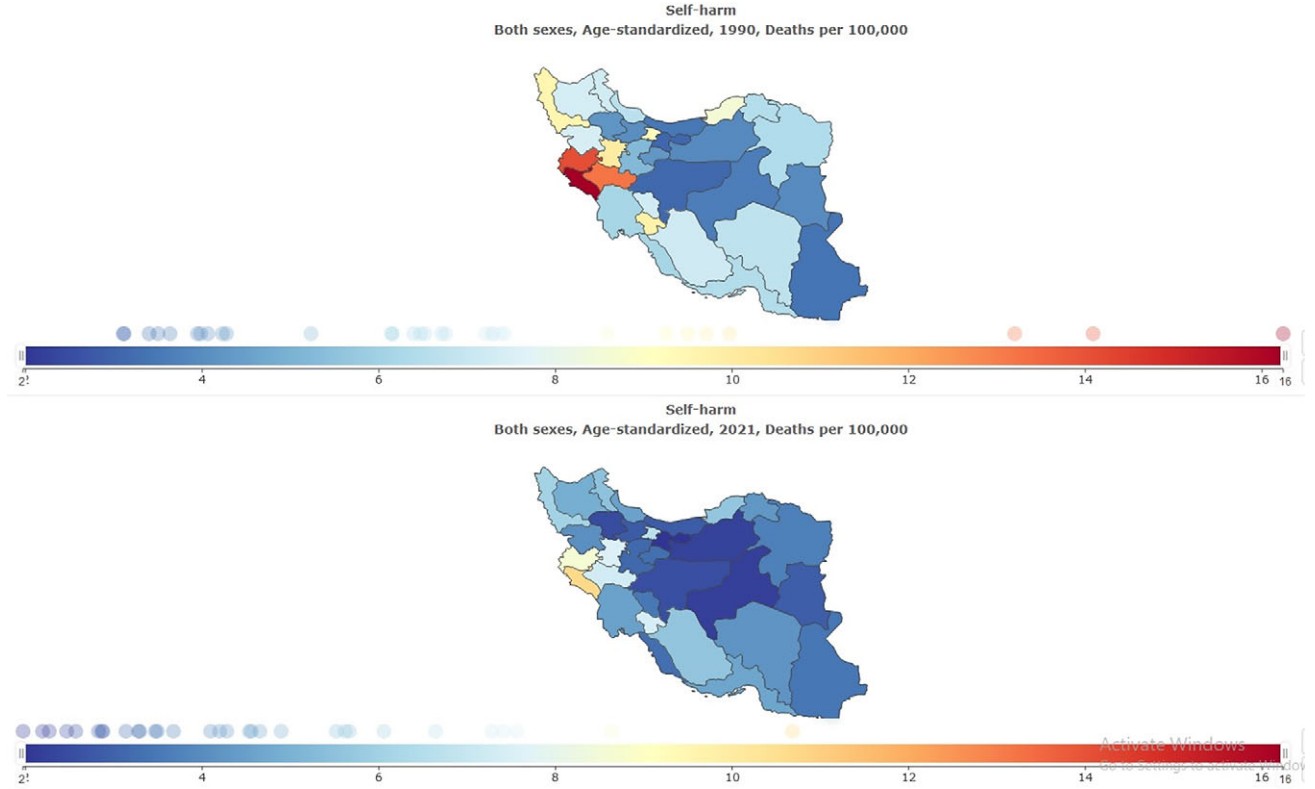

**Figure 11.** Age-standardized suicide mortality rate (per 100,000) in Iran, stratified by provinces, 1990–2021.

especially mental health-related care. Effective and efficient measures were taken to increase health and health care, but in the area related to mental health care, access to mental health services did not grow proportionately (Farzadfar et al., 2022), and this was one of the factors that affected mental health.

On the other hand, the stigma related to mental disorders is one of the issues that, although it has faded over time, still stands; therefore, the request related to mental health care has always been associated with stigma (Corrigan et al., 2014; Taghva et al., 2017). Another effective factor in mental disorders in Iran during recent decades is due to economic turmoil. Studies have shown that mental disorders, as well as self-harm and suicide, increase as a result of economic crises (World Health Organization. Regional Office for E., 2011; Marazziti et al., 2021). Some of the consequences of economic crises, such as unemployment, poverty and hyperinflation, are associated with an increase in mental disorders, such as depression and suicide, and disrupt mental health (Paul and Moser, 2009; Amiri, 2022a, 2022b).

A part of the economic turmoil was caused by the sanctions imposed on Iran (Kokabisaghi, 2018), which intensified especially since 2011. Although the factors affecting mental disorders, especially self-harm and suicide mortality, have had cumulative effects in Iran over the past decades, unlike depression and anxiety disorders, the prevalence of self-harm and suicide mortality is low in Iran, and it is lower than the world average and many countries with a high level of social welfare. The dominant religion of Iran is Islam, and in most Islamic countries, the suicide rate is low (Pritchard and Amanullah, 2007; Mirhashemi et al., 2016; Lew et al., 2022). This could be partly due to the role of religion. In Islam, suicide and self-harm are prohibited, and the sanctity of life is emphasized in Islamic teachings (Ali, 2000). Of course, less access to mental health care alone is not the determining factor. There is a stigma attached to suicide for families (Wyllie et al., 2025). The stigma associated with suicide is important in Iranian culture (Masoomi et al., 2022). One significant obstacle to accessing mental health services in Iran is the social stigma surrounding psychiatric issues (Taghva et al., 2017). Research conducted in Iran indicates that women often attempt to conceal their suicide attempts due to the fear of being stigmatized by society (Azizpour et al., 2018), and the burden of this stigma causes many individuals and families to not seek mental health care despite the need for mental health professionals and mental health care providers (Tay et al., 2018). Suicide is heavily stigmatized in predominantly Muslim countries like Iran, and similarly, other religious communities are not immune to this issue (Shoib et al., 2022).

The differences between sexes in self-harm and suicide mortality have consistently drawn the attention of researchers. Studies exploring these disparities reveal that suicide mortality is notably higher among males compared to females. Data published by the WHO further supports this finding, highlighting that, on a global scale, the age-standardized suicide rate for males is 2.3 times greater than that of females (World Health Organization, 2021). Major depressive disorder constitutes a significant factor contributing to suicide. While its prevalence is observed to be twice as high among females compared to males, the likelihood of suicide among females is markedly lower, amounting to only one-fourth of the probability seen in males (Murphy, 1998). The higher incidence of suicide among men has been attributed to several factors, including contrasting behavioral and emotional tendencies between genders. Men often prioritize independence and decisiveness, perceiving the admission of needing assistance as a sign of weakness, which leads them to avoid seeking help. On the other hand, women tend to emphasize interdependence and are

more likely to seek support from friends and accept help readily, contributing to differing responses in times of emotional distress (Murphy, 1998).

This study represents the first comprehensive endeavor to analyze trends in self-harm and suicide in Iran since 1990, offering gender- and age-specific estimates. However, it is subject to certain limitations extensively discussed within the framework of the GBD study. Challenges include issues related to the quality and collection of primary data, as well as inconsistencies in data availability. Moreover, the risk factors assessed are constrained by the exclusion of various potentially significant risk factors and covariates. The impact of the coronavirus disease 2019 pandemic further complicates the interpretation and analysis of findings. A more detailed discussion on the methodological constraints inherent in the GBD study is available in other sources (Brauer et al., 2024; Ferrari et al., 2024). GBD estimates are often derived from modeled data, particularly in contexts where the quality of surveillance systems is insufficient or constrained

## Health implications

The gathered data highlights the trends of suicide and suicide mortality in Iran over the past three decades. Despite notable advancements in physical health within the Iranian healthcare system during this period, demographic shifts and lifestyle changes have exposed gaps in addressing mental health. Efforts to improve awareness around mental health and ensure broader access to mental health services have remained insufficient. To address these challenges effectively, it is crucial to prioritize mental health literacy and expand access to care as key components in health policy development.

## Conclusion

The present study showed that despite demographic changes, economic turmoil and war, self-harm and suicide mortality have had a decreasing trend in Iran. Suicide mortality in males was higher than in females, self-harm was more common in females in past decades, and in recent years, it was lower than in males.

**Open peer review.** To view the open peer review materials for this article, please visit http://doi.org/10.1017/gmh.2025.10104.

**Abbreviations**

| | |
|---|---|
| DALYs | disability-adjusted life years |
| GBD | global burden of disease, encapsulating a comprehensive assessment of health losses due to diseases and injuries worldwide |
| ICD | International Classification of Diseases, providing a standard framework for categorizing health conditions |
| SEV | summary exposure value, quantifying exposure levels to specific risk factors |
| YLDs | years lived with a disability |
| YLLs | years of life lost due to premature death |

**Data availability statement.** The data sources of this study were taken from GBD 2021, which is publicly available. In this way, the data can be accessed through the links below: https://vizhub.healthdata.org/gbd-results/ https://vizhub.healthdata.org/gbd-compare/

**Acknowledgments.** The data sources of this study were taken from GBD 2021, which is publicly available.

**Author contribution.** Sohrab Amiri: Conceptualization, extraction, analysis, writing and revision.
Jannat Mashayekhi: Writing and revision.

**Financial support.** The study received no financial support.

**Competing interests.** The authors declare none.

**Ethics statement.** The data sources of this study were taken from GBD 2021, which is publicly available.

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
