## [Reviewer Report]

Percentage change was use for establishing trend. While percentage change is great for doing the trend in the data. I encourage adding a moving average as the data seem to be noisy. There were instance that the percentage change was negative and some positive. The trends in the data seems to have a lot of noise even though it is from the Global survey.

If this comment is not deemed relevant, will accept the paper as it is still well discussed.

---

## [Reviewer Report]

The manuscript is very important and timely, as there is more attention to the issue of mental health and prevention of suicide, globally and also in the area. The GBD 2021 data were employed to analyze long-term trends and estimated national and subnational burden of self-harm and suicide in Iran, it is a solid foundation for future studies.

The authors succinctly point to important trends, such as an overall reduction in the prevalence of self-harm across each time period (31 years) and gender inequalities in the rates of both self-harm and suicide This manuscript does a good job of identifying some of the clear findings such as the decline in the prevalence of self-harm across the 31-y time frame and enduring disparities in both the rates of self-harm and suicide mortality by gender. Sex, age and.isSuccessful.Patient sex, age and

location adds significant depth to the analysis and allows for more targeted interpretation. However, I believe the paper could be further strengthened by elaborating on some contextual factors that may have influenced these trends, such as changes in health service availability, sociopolitical developments, stigma, or policy shifts in Iran.

The observed increase in absolute suicide deaths, despite a reduction in age-standardized rates, is an important point that reflects population growth and demographic transitions. Including more interpretation of this dynamic would add value to the discussion.

Overall, the manuscript makes a valuable contribution to the literature on mental health and mortality in the Eastern Mediterranean region. With a few additions related to contextual interpretation and policy implications, the paper would be even more impactful for public health professionals and decision-makers.

---

## [Reviewer Report]

Line 34, 35: As the population increases, the need for mental health care will also increase.

Who said this, what made you say this, while suicide is mental health related indicator of a country, the quick assumption that increasing population has an increase ot the needs of mental health care is too quick to jump without reference to the data. Also the data is nto even talking about mental health care needs. The article premise may be coming from suicide population but it should not be said in an academic article. This may be said in an opinion piece, but in an academic article, it is best to cite or try to make an appropriate and coherent discussion as to why you say this. Because after this line you proceed to say that the population needs will also increase, indicating education, employment, welfare service etc. While there are papers highlighting its relationship, your paper just bluntly says it. It would be best to cite at this point. Also, that’s not your topic.

I’m not sold out on the highlighted part. it feels an opinion piece rather than an academic discussion on the implications of the data. It gives an idea what could be its effect and cause but it feels lacking, maybe an article specifying this things you’ve said.

In addition, the highlighted part again, "Of course, less access to mental

health care alone is not the determining factor, the stigma associated with mental

disorders is very important in Iranian culture, and the burden of this stigma causes

many individuals and families to not seek mental health care despite the need for

mental health professionals and mental health care providers. Especially in cases of suicide, many families try to hide the cause of death to avoid the stigma it

causes"

what made you say this, was there a qualitative study that explored those cases of suicide where the family had stigma about it and failed to seek help or assistance with regards to it.

I agree that this things are true but the thing is, it feels more of an opinion rather than an academic discussion. To improve it, I suggest find papers or articles that explore these things you have said and this would allow you explain it properly and have you backed up.

---

## [Editor Report]

The interpretation of demographic effects is important, especially the observation that absolute suicide deaths increased despite declining age-standardized rates, due to population growth and demographic transition. This point deserves deeper quantitative illustration, for example by presenting population growth alongside mortality trends in a supplementary figure. The discussion mentions war, economic turmoil, sanctions, and stigma, but these factors could be integrated more systematically, perhaps in a subsection linking major socio-political events to observed changes in self-harm and suicide. Although the introduction references global suicide rates, the discussion could benefit from direct comparison with other Middle Eastern countries to clarify whether Iran’s trends are unique or consistent with regional patterns. While the manuscript follows the GBD protocol, readers unfamiliar with the framework may struggle with terminology such as YLDs, YLLs, and DALYs. A brief paragraph simplifying these terms would improve accessibility.

The results are thorough but at times repetitive. Multiple sections restate that age-standardized rates decreased while absolute counts increased. This point could be consolidated to avoid redundancy. Figures are informative but could be enhanced with clearer labeling of years, uncertainty intervals, and inflection points such as the 2005 peak in incidence. Tables 5–7 on sex- and age-specific differences could be more impactful if the main message was summarized graphically, for example with a ratio of male-to-female suicide mortality over time.

The discussion section situates findings in historical and cultural context but could be more structured. It would help to first summarize key findings, then provide interpretation in relation to demographics, conflict, and economic pressures, and finally highlight policy implications. The role of religion is mentioned briefly but could be supported with empirical studies. The limitations section should more explicitly acknowledge that GBD estimates rely on modeled data in contexts where surveillance quality is limited.

The abstract could state more clearly the public health significance of findings. Currently it emphasizes descriptive trends but does not fully connect to policy relevance. The impact statement repeats several elements of the abstract and should instead focus on how the findings could inform interventions or health system planning. References need careful correction as some are inconsistently formatted, for example “Bank, T. W. (2 .(021GDP per capita …”. Multiple WHO references list “Regional Office for the Eastern” rather than “World Health Organization. Regional Office for the Eastern Mediterranean.” Figure captions should be expanded to include sample size, data source, and a one-sentence interpretation.

The manuscript is understandable but requires substantial language polishing for clarity and professionalism. Grammar and syntax should be corrected, for example “The number of suicide mortality in 2021 was 3,708 individuals” should be “The number of suicide deaths in 2021 was 3,708,” and “percentage change from 1990-2021 was -0.25” should be “The percentage change from 1990 to 2021 was -25%.” Repetition should be reduced, for example the phrase “prevalence, incidence, disability, and mortality rates” is overused. Typos include “compeletly teransperant” instead of “completely transparent” and “Demographic translation” instead of “demographic transition.” Some references are truncated or inconsistently capitalized. Formatting errors such as missing spaces before parentheses (for example “world(Knipe…”) should be fixed. Many sentences are too long and should be split for clarity, for example the sentence on population increase and suicide deaths in the discussion could be shortened into two simpler sentences.

---

## [Reviewer Report]

line 127 and 128 “This manuscript was produced as part of the GBD Collaborator Network and following the GBD Protocol.”

It would make the paper better if this summarized rather than referred.

Case definitions (line 142 to 152)

In the title, it states self-harm and suicide, however, in the case definition, it lacks definition on what suicide is and how suicide and self-harm differentiates from each other. To improve, I suggest adding discussion on its difference in the case definition.

This is great work; I look forward to having it done published. Iran’s mental health (suicide and self-harm statistics) trend has to be in the journals. Goodluck and keep it up! You’re doing a great work!